



**Contributions to OH reactivity from unexplored volatile organic compounds measured by**
**PTR-ToF-MS– A case study in a suburban forest of the Seoul Metropolitan Area during**
**KORUS-AQ 2016**
Dianne Sanchez,[1] Roger Seco,[1*] Dasa Gu,[1] Alex Guenther,[1] John Mak,[2] Youngjae Lee,[3] Danbi
Kim,[3] Joonyoung Ahn,[3] Don Blake,[4] Scott Herndon,[5] Daun Jeong,[1] John T. Sullivan,[6] Thomas
Mcgee,[6] and Saewung Kim[1*]
1. Department of Earth System Science, University of California, Irvine, Irvine CA 92697,
U.S.A.
2. School of Marine and Atmospheric Sciences, Stony Brooke University, Stony Brook, NY
11794, U.S.A.
3. National Institute of Environmental Research, Inchoen 22689, South Korea
4. Department of Chemistry, University of California, Irvine, Irvine CA 92697, U.S.A.
5. Aerodyne Research Inc., Billerica MA 01821, U.S.A.
6. NASA Goddard Space Flight Center, Chemistry and Dynamics Laboratory, Greenbelt, MD
20771, U.S.A.
* Now at: Terrestrial Ecology Section, Department of Biology, University of Copenhagen,
Copenhagen, Denmark and
Center for Permafrost (CENPERM), Department of Geosciences and Natural Resource
Management , University of Copenhagen, Copenhagen, Denmark
Corresponding author: saewung.kim@uci.edu, tel 1-949-824-4531
To be submitted to Atmospheric Chemistry and Physics





**Abstract**

We report OH reactivity observations by a chemical ionization mass spectrometer –

comparative reactivity method (CIMS-CRM) instrument in a suburban forest of the Seoul
Metropolitan Area (SMA) during Korea US Air Quality Study (KORUS-AQ 2016) from mid-
May to mid-June of 2016. A comprehensive observational suite was deployed to quantify
reactive trace gases inside of the forest canopy including a high-resolution proton transfer
reaction time of flight mass spectrometer (PTR-ToF-MS). An average OH reactivity of $30.7 \pm$
$5.1 \text{ s}^{-1}$ was observed, while the OH reactivity calculated from CO, NO + $NO_2$ ($NO_x$), ozone ($O_3$),
sulfur dioxide ($SO_2$), and 14 volatile organic compounds (VOCs) was $11.8 \pm 1.0 \text{ s}^{-1}$. An analysis
of 346 peaks from the PTR-ToF-MS accounted for an additional $6.0 \pm 2.2 \text{ s}^{-1}$ of the total
measured OH reactivity, leaving 42.0 % missing OH reactivity. The missing OH reactivity most
likely comes from VOC oxidation products of both biogenic and anthropogenic origin.












## 1. Introduction


50  Total OH reactivity ($s^{-1}$), the inverse of OH lifetime, is a measure of the total amount of

51 reactive trace gases in the atmosphere in the scale of reactivity, which allow us to quantitatively

52 evaluate our ability to constrain trace gases by comparing measurements of total OH reactivity

53 with the OH reactivity calculated from a speciated reactive gas measurement dataset. The

54 fraction of observed OH reactivity that cannot be reconciled by calculated OH reactivity is

55 known as "missing OH reactivity" (Di Carlo et al., 2004;Goldstein and Galbally, 2007;Yang et

56 al., 2016).  A substantial amount of missing OH reactivity has consistently been reported in

57 forest environments (30 - 80%). Di Carlo et al. (2004) conducted a study in a mixed forest near

58 Pellston, Michigan where they reported missing OH reactivity larger than observational

59 uncertainty. The authors concluded that the missing sources of reactivity were primary biogenic

60 volatile organic compound (biogenic VOC, BVOC) emissions, as the degree of missing OH

61 reactivity followed the temperature dependence of terpenoid emissions. In a boreal forest in

62 Hyytiälä, Finland, Sinha et al. (2010) report a similar result with observed trace gases that

63 account for only 50% of the measured OH reactivity. They argued that oxidation products of

64 BVOCs alone could not account for the missing OH reactivity. Thus, they also concluded that

65 primary emissions were more likely to be the source of missing OH reactivity and they further

66 suggest that this could be the result of the contribution of small amounts of many reactive gases.

67  On the other hand, some studies have attributed the sources of the missing OH reactivity

68 to unmeasured oxidation products of well-characterized BVOCs. Edwards et al. (2013)

69 measured OH reactivity in a pristine tropical forest in the Sabah region of Borneo during the

70 Oxidant and Particle Photochemical Processes (OP3) field campaign (Hewitt et al., 2010). This

71 study implemented the Master Chemical Mechanism (MCMv3.2) (Saunders et al., 2003;Jenkin





et al., 1997) into a box model framework to quantify potential contributions from unmeasured
oxidation products. The model was constrained with VOCs such as isoprene, monoterpenes, and
alkanes and alkenes and other observed trace gases such as $NO + NO_2$ ($NO_x$) and ozone ($O_3$).
The authors reported that the model simulated oxygenated VOCs (OVOCs) could contribute
47.1% of the calculated OH reactivity – surpassing the contribution from isoprene, the primary
emission of this ecosystem. It is notable that 30% of observed OH reactivity could not be
accounted for by the box model simulations. After examining the comprehensive observational
suite of VOCs, the authors determined that the most significant missing sources of OH reactivity
were likely secondary multifunctional carbon compounds rather than primary BVOC emissions.
Hansen et al. (2014) suggested that their observed missing OH reactivity were likely from
unmeasured oxidation products during the Community Atmosphere-Biosphere INteraction
EXperiment (CABINEX 2009) in Michigan. This notion was also consistent with findings
reported by Kim et al. (2011) who measured OH reactivity of branch enclosures from four
representative tree species in the forest canopy during the CABINEX study. They reconciled
most of the measured OH reactivity of four representative tree species with well-known BVOCs,
such as isoprene and monoterpenes. Finally, Nakashima et al. (2014) reported that 29.5% OH
reactivity could not be reconciled by the speciated trace gas dataset during the Bio-hydro-
atmosphere interactions of Energy, Aerosols, Carbon, $H_2O$, Organics and Nitrogen-Southern
Rocky Mountain 2008 (BEACHON-SMR08) field campaign (Ortega et al., 2014). The campaign
took place at the Manitou Experimental Forest (MEF) in Colorado, a ponderosa pine plantation
dominated by primary BVOC emissions of 2-methyl-3-butene-2-ol (232-MBO) and
monoterpenes (Ortega et al., 2014). The authors also reported that the missing OH reactivity was
likely from BVOC oxidation products. In the same context, Kim et al. (2010) conducted PTR-





MS mass spectrum analysis for both ambient air and branch enclosures at the MEF site. They
reported more conspicuous unidentified signals on PTR-MS mass spectra in the ambient samples
than those from branch enclosure samples at this site.
During the Southern Oxidant and Aerosol Study (SOAS) in 2013, Kaiser et al. (2016)
used a comprehensive suite of VOC measurements at an isoprene dominant forest site in the
southeastern US to examine the role of the OVOC species in missing reactive carbon. The
authors used MCMv3.2 embedded in the University of Washington Chemical Box Model
(UWCM) to compare OH reactivity from model-generated OVOCs to OH reactivity from
measurements of OVOCs. There was no significant discrepancy between the average measured
and calculated OH reactivity including observed trace gases and model calculated oxidation
products of VOCs. However, it was noted that a small portion ($1 \text{ s}^{-1}$) of observed OH reactivity
could not be reconciled by the model calculation. As this fraction was not correlated to isoprene
oxidation products, it was suggested that the missing OH reactivity may be due to unmeasured
primary emissions. One caveat of this analysis pointed out by the authors was that the
concentrations of the modeled first-generation isoprene oxidation products (e.g. MVK, MACR,
isoprene hydroxy hydroperoxides (ISOPOOH), isoprene nitrates (ISOPN), and hydroperoxy
aldehydes (HPALD)) were significantly overpredicted in the afternoon. Consequently, the
uncertainty of the model calculation is likely to be much higher for the multi-generation
oxidation products and their contributions to the OH reactivity contributions. This result
highlights the uncertainty in relying solely on box-model results to assess OH reactivity.
This study examines the OH reactivity observations at Taehwa Research Forest (TRF)
supersite from 15 May 2016 to 7 June 2016 during the Korea United States Air Quality Study
2016 (KORUS-AQ 2016) campaign. TRF (37 18' 19.08" N 127 19' 7.12" E, 162 m altitude) is





operated by Seoul National University and located in Gwangju in the Gyunggi Province in South
Korea (Kim et al., 2013b). The site is about 35 km southeast from the center of Seoul and
borders the greater Seoul Metropolitan Area (SMA) with its population of 25.6 million. This
geographical proximity to SMA results in a significant level of anthropogenic influence,
particularly in elevated $NO_x$ (Kim et al., 2016). Additionally, occasional pollution transport
events occur at regional scales. Previous studies at the site have consistently highlighted the
importance of BVOC photochemistry at TRF (Kim et al., 2016;Kim et al., 2013a;Kim et al.,
2015).  Isoprene and monoterpenes are the dominant OH sinks at the site among observed VOCs.
The elevated $NO_X$ accelerates the photochemical processing of VOCs (Kim et al., 2015). Thus,
this site is an ideal natural laboratory to study contributions towards total OH reactivity from
primary trace gas emissions from both natural and anthropogenic processes and their oxidation
products. This motivated us to deploy a high-resolution proton transfer reaction time-of-flight
mass spectrometer (PTR-ToF-MS) to quantify trace amounts of VOCs with unknown molecular
structures by taking advantage of the universal sensitivity of hydronium ion chemistry towards
reactive VOCs (Graus et al., 2010;Jordan et al., 2009a). Therefore, we intend to observationally
constrain the contributions of conventionally unidentified or unmeasured VOCs towards OH
reactivity.

**2. Methods**
***2.1. Field Site***

The Taehwa Research Forest is a Korean pine (*Pinus koraiensis*) plantation (300 m × 300

m) surrounded by a deciduous forest dominated by oak trees (Kim et al., 2013b). A flux tower
(40 m height) at the center of TRF has air-sampling inlets at multiple heights (4 m, 8 m 12 m,



and 16 m) below the canopy top (20 m). Each inlet consists of Teflon tubing (3/8" OD) with ~ 1
second of residence time. The trace gas dataset including VOCs presented is the average of
concentrations measured at the inlets inside of the canopy as previous studies illustrate that there
is no substantial vertical VOC gradients inside of the canopy (within 3 %, Kim et al. (2013b)).
An air-conditioned instrument shack located at the base of the flux tower housed the PTR-ToF-
MS for VOC measurements, a mini tunable infrared laser direct absorption spectroscopy (mini-
TILDAS) instrument for HCHO, methane, and methanol measurements, and analyzers for
carbon monoxide (CO), sulfur dioxide ($SO_2$), ozone ($O_3$), and meteorological measurements. The
OH reactivity and $NO_x$ analyzers were located in another nearby air-conditioned shack (3 m
apart) and sampled air through an extended Teflon inlet line of 4 m (¼" OD) from the ground
with a flow rate of 4 sLpm resulting in a 0.5 second residence time. The analytical characteristics
of the instrumentation suite are summarized in Table 1. A ceilometer backscattering
characterized boundary layer vertical structure at the site. The ceilometer analysis described by
Sullivan et al. (2019) reveals the diurnal boundary layer height evolution, indicating a maximum
in the afternoon around 1-3 km and a minimum in the early morning below 500 m.

### 2.2. OH Reactivity Measurements

A chemical ionization mass spectrometer – comparative reactivity method (CIMS-CRM)
instrument was used to measure OH reactivity. The UCI CIMS-CRM system includes a chemical
ionization mass spectrometer with a hydronium reagent ion. The CRM method measures total
OH reactivity by quantifying the relative loss of pyrrole, a highly reactive gas ($k_{OH+ pyrrole} = 1.45$
$\times 10^{-10}$ $cm^3$ $molecule^{-1}$ $s^{-1}$ at 298 K) that is rarely found in the atmosphere (Sinha et al., 2008b).
Nitrogen gas flows through a bubbler full of ultrapure liquid chromatography mass spectrometer



(LC-MS) grade water to produce water vapor. The water vapor then flows into a glass reactor
where it is photolyzed into OH radicals by a mercury lamp (Pen-Ray® Light Source P/N 90-
0012-01). The measurement uncertainty is 16.7% (1σ) with a limit of detection of 4.5 s$^{-1}$ over 2
minutes (3σ).

The UCI CIMS-CRM instrument has been deployed on multiple occasions, including the

Megacity Air Pollution Study (MAPS)-Seoul 2015 campaign that incorporated previous
measurements at the TRF ground site during September 2015 (Sanchez et al., 2018;Kim et al.,
2016). During the SOAS 2013 campaign, an ambient OH reactivity intercomparison study was
conducted with laser induced fluorescence (LIF) system (Sanchez et al., 2018). The instrument
intercomparison showed that the OH reactivity measurements from the CRM and LIF
instruments generally agreed within the analytical uncertainty. An average of 16% difference
between the techniques was noted in the late afternoons where the CRM measurements were
lower than those from LIF. As discussed in Sanchez et al. (2018), this is likely caused by the
difference in sampling strategies, as the CRM measurements relied on a lengthy Teflon inlet (15
m) while the LIF directly sampled air at the top of a walk up tower. As mentioned above, at TRF
we used a shorter inlet line to minimize residence time and avoid inlet line loss.

An extensive intercomparison study was conducted by Fuchs et al. (2017) with various

OH reactivity measurement techniques that highlighted potential analytical artifacts in the CRM
technique. These artifacts have all been examined and preventive measures have been
implemented in the UCI CIMS-CRM system deployed at TRF. This included a laboratory-built
catalytic converter (Pt-wool at 350 °C) that minimized the interferences due to changes in air to
prevent the interference from the difference in humidity for the zero air characterizations.
Additionally, interference from the OH recycling reaction of HO$_2$ with NO in the glass chamber





was prevented by removing OH reactivity data points that corresponded with ambient NO levels
that exceeded 5 ppb to reflect results from a laboratory characterization study (Sanchez et al.,
2018). Finally, we performed multi-point calibrations with a propene mixture using a NIST
traceable gas standard (AirLiquide LLC, 0.847 ppm) during the field campaign to avoid any
circumstances where the pseudo first-order reaction regime is not established. Detailed
calibration procedures for the OH reactivity system including laboratory multi-component
calibration results can be found in Sanchez et al. (2018).

***2.3.PTR-ToF-MS Measurements***

A high-resolution PTR-TOF-MS (Ionicon Analytik GmbH) (de Gouw and Warneke,

2007)· (Jordan et al., 2009b) was deployed at the TRF site. The instrument was operated with a
drift tube temperature of 60 ℃, 560 V drift voltage, and 2.27 mbar drift tube to maintain E/N of
126 Td. Background checks were manually conducted about three times a day for a 10-minute
duration by scrubbing the ambient air through a catalytic convertor (Pt-wool maintained at
350°C). The detectable peaks from the ambient spectra were assessed by subtracting the
background spectrum. The instrument was calibrated with a gas mixture manufactured by Apel-
Riemer Environmental Inc. The mixture contains ~ 1 ppmv of acetaldehyde, acetone, isoprene,
methyl vinyl ketone, methacrolein, benzene, methly ethyl ketone, toluene, o-xylene, and α-
pinene.  The concentration of the compounds were assessed in the Blake Lab at University of
California, Irvine, who also conducted the airborne VOC analysis using whole air samples
during the KORUS-AQ campaign on the NASA DC-8 (Colman et al., 2001).

A mass range of m/z 40 to m/z 267 was analyzed from the recorded PTR-ToF-MS mass

spectra. An automatic mass scale calibration was conducted every 5 minutes on the data



averaged over 30 seconds. The raw PTR-ToF-MS data were processed using the PTRwid
software described by Holzinger (2015). We normalized the mass peaks by $10^6$ reagent ion
counts ($H_3O^+$). As the majority of the VOC mass peaks could not be directly calibrated, we
determined the VOC sensitivities using equation 1 (Eq 1). We obtained a VOC mixing ratio
($ppb_{VOC}$) by multiplying mass discrimination corrected normalized counts for each VOC
($ncps_{VOC}$) by their proton transfer reaction rate coefficient value ($k_{VOC}$) (Cappellin et al., 2012).
The benzene calibration factor was used to calculate mixing ratios by applying its proton transfer
reaction rate coefficient ($k_{benzene}$) and sensitivity (ncps ppb$^{-1}$) for the available compounds.

$$ppb_{VOC} = \frac{ncps_{VOC}}{11.94\frac{ncps_{benzene}}{ppb}} \times \frac{k_{benzene}}{k_{VOC}}$$
        Eq. 1


For the mass peaks where specific proton transfer reaction rates were unavailable, we estimated
the mixing ratios by applying a proton transfer reaction rate coefficient ($k_{H3O+}$) of $3.00 \times 10^{-9}$ cm$^3$
s$^{-1}$, the default value for PTRwid calculations. The spectra had a limit of detection of tens of ppt
for a 30 second average. The calibrated compounds had a range of detection limits as low as 3.7
ppt for α-pinene and as high as 48 ppt for toluene.

**2.4. *OH Reactivity Calculation***
OH reactivity was calculated from the concentrations of all the compounds observed by
the instrumental suite described in Table 1. The original data can be found in the KORUS-AQ
2016 data archive at https://korus-aq.larc.nasa.gov/. A total of 360 mass peaks measured by the
PTR-ToF-MS were analyzed above the background (3 $\sigma$ or above) to assess their contribution to
the calculated OH reactivity. Fourteen of the mass peaks were identified as VOCs commonly



reported for PTR-MS measurements (Table 1), leaving 346 unidentified peaks. These remaining
mass peaks were grouped into three categories in order to estimate their possible OH reactivity
contribution.

Category I included mass peaks for which the PTRwid software calculated a molecular

formula. OH reaction rate coefficients for the individual peaks were obtained from the National
Institute for Standards and Technology (NIST) Webbook library. If the information was
unavailable from the NIST Webbook database, a structure-reactivity relationship described by
Kwok and Atkinson (1995) was applied to obtain reaction rate coefficients. This is an empirical
calculation system to estimate $k_{OH}$ based upon the number of carbons and the functional groups
of given VOCs. The framework is able to calculate $k_{OH}$ within a factor of two according to a
thorough assessments presented in Kwok and Atkinson (1995). However, the authors discourage
the application of the framework to compounds that were not examined in the study such as
halogenated compounds. Although halogenated compounds are not included in this study, one
should be aware of a potentially significant uncertainty.

Category II included mass peaks for which the PTRwid software could not assess an

exact molecular composition due to uncertainty in the data processing system. Nonetheless, this
group of compounds illustrated a positive correlation with either anthropogenic (benzene,
toluene) or biogenic (MVK+MACR and monoterpenes) VOCs.  Category II compounds are
further grouped into subcategories corresponding to these two main VOC sources.
OH reaction rate constants ($k_{OH}$) were estimated with equations based on the relationship
between the $m/z$ and the $k_{OH}$ of compounds in Table 1 (Figure S1). The compounds were grouped
into 5 $m/z$ bins and the average $k_{OH}$ of each bin was calculated. The green triangles represent 5
$m/z$ binned averages from these compounds plotted with their respective average $k_{OH}$. This


approach can be justified by the fact that the reaction constants of VOCs towards OH tend to
increase as a function of molecular mass within functional groups (Kwok and Atkinson,
1995;Atkinson, 1987).  The y-intercepts of the linear regressions were assessed using the $k_{OH}$
values of the biogenic or anthropogenic compounds and their masses.

Category III included mass peaks with very low mixing ratios (average = 4.8 ppt ±19.5

ppt) that were above the limit of detection. We applied a $k_{OH}$ corresponding to the dark green
best-fit line in Figure S1 to these peaks. The y-intercept of the dark green line was based on that
of acetaldehyde, as it was the lowest mass compound used for the OH reactivity calculations in
this study.

**3.  Results and Discussion**

An average OH reactivity of $30.7 \pm 5.1$ s$^{-1}$ was observed from 15 May – 7 June 2016

(Figure 1). This was within the range of OH reactivity observed in urban regions (10 - 33 s$^{-1}$).
(Kovacs et al., 2003;Ren et al., 2003;Sinha et al., 2008a;Dolgorouky et al., 2012;Whalley et al.,
2016;Kim et al., 2016;Yang et al., 2017)  and in the range of previously reported observations
and model calculations at the TRF site (~15 - 35 s$^{-1}$) (Kim et al., 2016;Kim et al., 2015). The
total calculated OH reactivity of $11.8 \pm 1.0$ s$^{-1}$ from the measured compounds in Table 1 resulted
in 63.3% missing OH reactivity. However, an additional OH reactivity of $6.0 \pm 2.2$ s$^{-1}$ was
further calculated from the reactivity of the VOCs in Categories I – III. The contribution lowered
the missing OH reactivity level to 42% of the measured OH reactivity. Kim et al. (2016) had
previously measured an average OH reactivity of 16.5 s$^{-1}$ at TRF during the MAPS-Seoul
campaign from 1 September – 15 September 2015, a substantially lower level then what we
report during this springtime study. Although small alkanes and alkenes such as ethane, ethene,



propane and propene were not observed on the site, we utilized the dataset from the NASA DC-8
that flew at 700 m above the site, which indicates that their contribution was consistently small
($\sim 0.7$ s$^{-1}$ in average).

The difference can be attributed to the notably higher reactive trace gas loadings during

KORUS-AQ compared to the TRF measurements during MAPS-Seoul. The NO$_x$, benzene, and
toluene concentrations were 3 times higher during KORUS-AQ and CO was 1.4 times higher
(Figure S2). Although the average isoprene concentrations were similar between the two
campaigns, MVK and MACR concentrations during KORUS-AQ were ~3 times higher,
illustrating a higher oxidative environment. There was a persistently high MVK+MACR to
isoprene ratio of 1.8 during the KORUS-AQ campaign at TRF. This ratio was similar to the
value reported during the summer in a moderately polluted forest in the Pearl River Delta that
was attributed to a strong atmospheric oxidation capacity (Gong et al., 2018). The missing OH
reactivity during KORUS-AQ was generally much higher than levels reported during urban
observations (up to 50% missing OH reactivity) (Kovacs et al., 2003;Ren et al., 2003;Sinha et
al., 2008a;Dolgorouky et al., 2012;Whalley et al., 2016;Kim et al., 2016;Yang et al., 2017) and
within the range of previously reported values in forest regions where as much as 80% missing
OH reactivity has been reported (Kim et al., 2016;Di Carlo et al., 2004;Nolscher et al.,
2012;Edwards et al., 2013;Nolscher et al., 2016;Ramasamy et al., 2018;Nakashima et al., 2014).

Figure 2 shows the diurnal average of measured, calculated, and missing OH reactivity

from 15 May – 7 June 2016. Isoprene was the largest contributor to VOC OH reactivity in the
afternoon and the early evening (36% of the calculated OH reactivity in the evening), consistent
with the previous studies conducted in this site (Kim et al., 2016;Kim et al., 2013b;Kim et al.,
2015). Among all the trace gases, the largest average contributor to the calculated OH reactivity



was $NO_x$, which contributed 18.2% (5.6 s⁻¹) to the measured OH reactivity. The $NO_x$
contribution to OH reactivity is higher during the morning and evening rush hours and at a
minimum in the afternoon, which has been reported consistently in previous reports conducted
near megacities (Kovacs et al., 2003;Mao et al., 2010;Dolgorouky et al., 2012;Ren et al.,
2003;Shirley et al., 2006).  Enhanced OH reactivity during the morning or night and minimum
OH reactivity during the afternoon have been reported in urban areas (Kovacs et al., 2003;Ren et
al., 2006;Shirley et al., 2006;Dolgorouky et al., 2012;Mao et al., 2010;Whalley et al., 2016). On
the other hand, strong light-sensitive biogenic emissions (e.g. isoprene) result in a maximum
observed OH reactivity in the afternoon in forested regions (Ren et al., 2006;Sinha et al.,
2012;Edwards et al., 2013;Hansen et al., 2014;Zannoni et al., 2017;Nolscher et al., 2016) . One
exception is an OH reactivity observation conducted in Hyytiälä, a forested site that has low
isoprene levels, by Sinha et al. (2010). They attributed a flat diurnal OH reactivity variation to
the interplay between high daytime emissions and low nighttime boundary layer height. In urban
environments, it is mostly anthropogenic trace gases such as aromatics and OVOCs that
contribute to OH reactivity. These compounds have a longer lifetime compared to the diurnal
boundary layer evolution. This leads to the accumulation of such compounds in the shallow
boundary layer during the night. On the other hand, strong emissions of reactive BVOCs in
deciduous forest regions enhance OH reactivity during the daytime but then quickly react away.
Very subtle diurnal differences observed in this study (Figure 2), therefore, can be understood as
the competitive influences of both anthropogenic and biogenic compounds to the OH reactivity.

As described in detail in Sullivan et al. (2019) and Jeong et al. (2019), a strong regional

stagnation episode occurred during the KORUS-AQ campaign between May 17 – 23. Later, the
Korean Peninsula was affected by a period of continental pollution outflow between May 28 and



June 1. The diurnal averages of the two periods and their calculated OH reactivity are presented
in Figure 3. It is notable that there is very little difference in the observed OH reactivity between
the two distinct periods in terms of the amount of OH reactivity and its diurnal pattern (Figure 4).
Furthermore, no significant variance of the different classes of reactive gases such as criteria air
pollutants (CO, $NO_x$, $O_3$, and $SO_2$), OVOCs (acetone, acetaldehyde, formaldehyde,
methylglyoxal, methanol, methyl ethyl ketone), aromatics (benzene, toluene, xylenes, styrene,
benzaldehyde, trimethylbenzenes), and BVOCs (isoprene, monoterpenes, sesquiterpenes,
MVK+MACR) was observed during the different periods (Figure 5). These different classes of
reactive gases generally differed by less than 10% during the two periods from the overall
campaign. This observation shows that the presence of reactive gases is mostly controlled by
relatively short-lived compounds determined by local emissions and their oxidation products.

The diurnal variation behavior of each chemical class reflects the chemical lifetime of the

compounds (e.g. aromatics vs BVOCs). The calculated OH reactivity from OVOCs does not
show a strong diurnal variation. This reflects the fact that OVOCs are mostly generated or
emitted during the daytime and their lifetime is generally longer than their precursors, which
allows nocturnal accumulation due to the absence of OH. The differences in the diurnal variation
of different classes of reactive gases can also be used to interpret the origin of the compounds in
Categories I-III as presented in Figure 6. The diurnal variations of Category I resemble those of
relatively long-lived chemical species with a distinct nocturnal accumulation pattern. This
diurnal pattern has been previously reported for both anthropogenic VOCs such as toluene and
benzene and temperature dependent monoterpenes such as $\alpha$-pinene. It is notable that the diurnal
pattern is enhanced during the stagnation period during early morning hours. This enhancement
is also seen in the aromatic trace gases particularly during the stagnation period (Figure 5b).


Indeed, there are both biogenic and anthropogenic contributions towards the Category I

compounds, which contribute an average of 3.8 $s^{-1}$ to the OH reactivity assessment, the largest

amount among the three categories (Figure 6a). The largest contributors to Category I, which

appear to be from a mixture of biogenic and anthropogenic sources, include *m/z* 89.060, 101.06,

and 101.096, and they contributed 0.3 $s^{-1}$, 0.2 $s^{-1}$, and 0.2 $s^{-1}$, respectively. The m/z 89.060 had a

molecular formula of $C_4H_8O_2H^+$ and was correlated to the anthropogenic compounds such as

benzene and toluene. The *m/z* 101.06 peak had the molecular formula of $C_5H_8O_2H^+$ and had a

diurnal variation similar to that of MVK + MACR. This mass peak has been previously

identified in laboratory (Zhao et al., 2004) and field (Williams et al., 2001) studies as the $C_5$

hydroxy carbonyl, an isoprene oxidation product. Results from an indoor chamber

photooxidation experiment conducted by Lee et al. (2006) showed that *m/z* 101 is a common

fragment of unidentified oxidation products of monoterpenes, sesquiterpenes, and isoprene. Lee

et al. (2006) also reported that this mass peak also composed over 5% of the fragments of

unidentified α- humulene and linalool oxidation products. The molecular formula of this peak is

$C_6H_{12}OH^+$, and it has been identified in previous studies as $C_6$ carbonyls (Koss et al., 2017) or

hexanal (Brilli et al., 2014;Rinne et al., 2005). Furthermore, *m/z* 99.044 and 113.023 were also

among the highest contributors to Category I and were correlated with MVK and MACR. The

*m/z* 99 was previously reported to be a fragment ion of unidentified terpene oxidation products in

a chamber experiment (Lee et al., 2006). The *m/z* 113 was observed by a PTR-MS in a

Ponderosa pine forest in central California by Holzinger et al. (2005). In this case, it was formed

within the canopy from the rapid oxidation of terpinolene, myrcene, and α-terpinene.

Furthermore, *m/z* 113 was observed to come from the photooxidation and ozonolyis of multiple

terpenes in two indoor chamber studies by Lee et al. (2006). The m/z 113 composed over 5% of



the oxidation product fragments of myrcene and verbenone. Finally, m/z 83.085 had the
molecular formula of $C_6H_{11}^+$ and was correlated to benzene. Multiple studies have identified this
peak as cyclohexane, methyl-cyclopentane, or methylcyclohexane, typically found in areas rich
in oil and gas (Koss et al., 2017;Gueneron et al., 2015;Yuan et al., 2014). In summary, both the
gross diurnal pattern and the individual peak analyses consistently illustrates that both
anthropogenic and biogenic compounds comprise Category I, the largest contributor to the
previously unexplored compounds in the PTR-ToF-MS spectrum at this research site.

Category II contributed an average of 0.3 $s^{-1}$ to the calculated OH reactivity, the lowest

amount for the three Categories (Figure 6b). The compounds in category II appear to correlate to
either BVOCs or acetone, depending on the time period. In Figure 6b, the maximum during the
transport period is enhanced to about 0.2 $s^{-1}$ higher than the overall campaign and shifted about 3
hours later to ~4:00 PM. The OH reactivity calculated from Category II is strongly correlated to
MVK + MACR ($r^2 = 0.82$) during this period as well. On the other hand, during the stagnation
period the average OH reactivity from Category II correlates more strongly with acetone ($r^2 =$
0.62) than with MVK +MACR ($r^2 = 0.28$). In fact, six of the highest contributors to Category II
(Figure 6b) are more strongly correlated to acetone ($r^2 > 0.40$) during the stagnation period
compared to the transport period. The sources of acetone can be either biogenic or
anthropogenic. Biogenic sources include direct emissions from plants or their oxidation products
and plant decay (Jacob et al., 2002;Seco et al., 2007). Anthropogenic sources of acetone include
vehicular emissions, solvent use, and the oxidation of other anthropogenic VOCs (Jacob et al.,
2002). Therefore, this illustrates that the compounds in Category II also have a complex source
profile of both biogenic and anthropogenic origin.





Category III contributed 1.9 s$^{-1}$ to the calculated OH reactivity (Figure 6c). The six
highest contributors out of 236 mass peaks contributed a total of 0.43 s$^{-1}$ of the calculated OH
reactivity. Overall, Category III compounds had no strong correlations to isoprene,
MVK+MACR, benzene, or toluene during either the stagnation or transport periods. However,
Category III compounds were highly correlated to methylglyoxal ($r^2$ = 0.85, 0.82, and 0.78 for
the stagnation, transport, and overall period, respectively), one of the measured OVOCs. A
global modeling study illustrated that methylglyoxal is mainly produced from isoprene oxidation
processes and the second most important source is acetone oxidation (Fu et al., 2008). In
addition, aromatics and alkenes are also known to produce methylglyoxal through atmospheric
oxidation processes (Henry et al., 2012). As TRF is a high aromatics and high isoprene
environment, the source profile of methyl glyoxal in the region is likely complex, which can be
applied to interpret the source of the Category III compounds.
Overall, the OH reactivity estimates from Categories I – III contributed an average of 6.0
± 2.2 s$^{-1}$ to the calculated OH reactivity. In summary, there is consistency that both
anthropogenic and the biogenic contributions need to be further studied in the PTR-ToF-MS
spectrum. Furthermore, by adding this additional signal from Category I, II, and III, VOC
contribution to calculated OH reactivity (11.0 s$^{-1}$) becomes larger than that (6.8 s$^{-1}$) from criteria
air pollutants (CO, NO$_x$, SO$_2$ and O$_3$). This should be considered when evaluating ozone
production regimes (Kim et al., 2018).
Even with the inclusion of the additional peaks to the calculated OH reactivity, we still
find a missing OH reactivity of 42%. Thus, it is important to investigate the origin of this
missing fraction. A correlation can be observed between missing OH reactivity in percentage and
OH reactivity from NO$_X$ (R2 = 0.5, Figure 7 A) but not between OH reactivity from NO$_x$ and



absolute missing OH reactivity ($s^{-1}$) (R2 = 0.2, Figure 7 B). This leads us to speculate that there
is a consistent presence of unquantified trace gases causing missing OH reactivity. As $NO_x$
illustrates a conspicuous temporal variation that appears to correlate with the fraction of missing
OH reactivity, while observed OH reactivity and calculated OH reactivity from VOCs indicate a
less pronounced diurnal difference.

Finally, unaccounted for uncertainty associated with the reaction rate constant

estimations described in the method section should be also further explored. For example, to
reconcile the averaged missing OH reactivity during the day (10 $s^{-1}$), it requires ~ 60 ppm of
methane but only ~ 4 ppb of isoprene. This clearly demonstrates the importance of rate constant
estimation. Indeed, if we apply the reaction rate constant of isoprene with OH ($k_{OH} = 1 \times 10^{-10}$
$cm^3$ molecule$^{-1}$ $s^{-1}$ at 298 K) to Category II and Category III compounds, then the observed OH
reactivity is fully reconciled (Figure S3). Proton ion chemistry may have an intrinsic limitation to
quantify highly oxidized OVOCs. Moreover, due to the different inlet configurations for OH
reactivity and VOC observations, their contributions towards observed and calculated OH
reactivity may not have been consistently evaluated. Therefore, a comprehensive analysis along
with a dataset from other instrumentation is necessary towards reconciling missing OH reactivity
with observational constraints. Finally, it is highly plausible that we may double count for
fragmented molecules in the mass spectrum. Although it would not affect concentration
evaluation as the intensity of ion signals from the fragmented molecules would be fully
accounted for by adding parent ion and fragmented ion signals, the OH reactivity calculated from
the fragmented ions is susceptible to underestimation from the assumption that $k_{OH}$ positively
correlates with molecular masses.





### *4. Summary*
We present OH reactivity observations at a suburban forest site during the KORUS-AQ field
campaign. A comprehensive trace gas dataset including 14 VOCs quantified by PTR-ToF-MS is
used to calculate OH reactivity, which only accounts for 36.7 % of the averaged observed OH
reactivity.
This study presents a detailed methodology for retrieving OH reactivity contributions from
all of the peaks of the PTR-ToF-MS mass spectrum. This decreases the amount of missing OH
reactivity as the majority of them have not been accounted towards calculated OH reactivity in
previous studies. First, we converted the raw signals to concentrations using a constant proton
transfer reaction rate ($3 \times 10^{-9}$ cm$^3$ s$^{-1}$). Then, we grouped the previously unaccounted peaks into
three categories to estimate reaction constants for each compound. The contributions of the
unaccounted peaks in the mass spectrum account for a calculated OH reactivity of $\sim 6$ s$^{-1}$, which
decreases missing OH reactivity from 63.3 % to 42.0 %. It is noteworthy that the diurnal
variations of observed OH reactivity and calculated OH reactivity from the various groups of
trace gases does not have a high variability during the field campaign even though there were
several synoptic meteorological configuration changes. This suggests that the reactive trace gas
loading is mostly determined by local emission and oxidation processes not influenced by the
synoptic meteorological conditions.
In conclusion, this study highlights PTR-ToF-MS as a tool for observationally constraining
missing OH reactivity. Further study is required particularly towards characterizing proton
reaction rate constants and reaction constants with OH for the many unknown compounds
detected on PTR-ToF-MS. In addition, other mass spectrometry techniques, such as nitrate or
iodine ion chemistry systems, should be utilized in future studies to complement the PTR



technique, which is sensitive to volatile to semi volatile VOCs, to quantify lower volatility
compounds and comprehensively constrain OH reactivity contributions from VOCs.

**Acknowledgements**

This study is supported by NASA (NNX15AT90G) and NIER. We highly appreciate
NASA ESPO for logistical support. Taehwa Research Forest is operated by College of
Agriculture and Life Sciences at Seoul National University.

**Data Availability**

Data is available at: https://korus-aq.larc.nasa.gov/

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

**Tables and Figures**

Table 1. Description of instrument and measured parameters.

| Instrument | Parameters | Measurement Uncertainty (1$\sigma$) |
|---|---|---|
| Chemical Ionization Spectroscopy - Comparative Reactivity Method - (CIMS-CRM) | OH reactivity | 16.7% |
| Thermo Scien8fic 42i | NO | 20% |
| Cavity Ring Down Spectroscopy | NO$_2$ | 20% |
| Thermo Scientific 49i | O$_3$ | 4% |
| Lufft 501 C | Temperature | ±0.3 °C |
| Thermo Scientific 48i TLE | CO | 10% |
| Thermo Scientific 43i TLE | SO$_2$ | 10% |


| | | |
|---|---|---|
| Mini Tunable Infrared Laser Direct Absorption Spectroscopy (mini-TILDAS) Formaldehyde Monitor | HCHO, CH$_4$, CH$_3$OH | 5% |
| Proton Transfer Reaction Time of Flight Mass Spectrometer (PTR-TOF-MS) | Acetaldehyde, Ethanol, Acetone, Isoprene, MVK + MACR, Methyl ethyl ketone, Benzene, Monoterpenes, Toluene, Furfural, Benzaldehyde, Xylenes, Trimethylbenzenes, Sesquiterpenes | Isoprene 9.8% Benzene 6.9% Toluene 6.5% Monoterpenes 9.2% Xylenes 4.0% Other 16.5% |













Figure 1. Observed and calculated OH reactivity during KORUS-AQ 2016. The measured and
calculated OH reactivity are on the left axis while the missing OH reactivity is on the right axis.
The yellow box represents the stagnation period and the blue box represents the transport period.

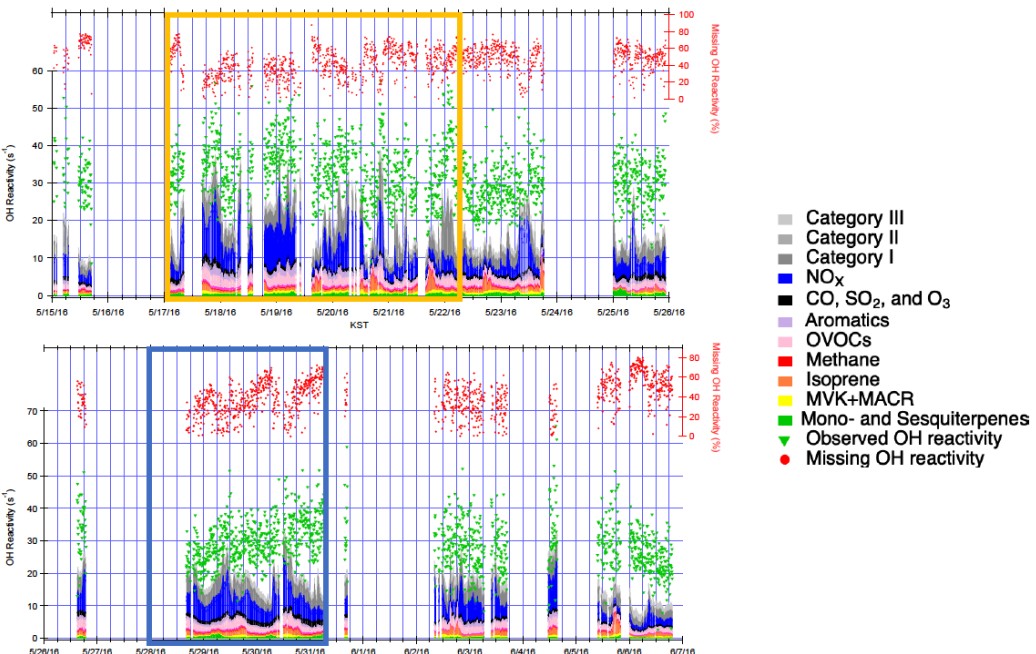







Figure 2. The diurnal average of OH reactivity from 15 May 2016 – 7 June 2016. The measured
and calculated OH reactivity are on the left axis. The blue shading represents uncertainty in the
measured OH reactivity. The black bars represent the propagated uncertainty of calculated OH
reactivity. The missing OH Reactivity in the percentage scale can be read using the right axis.

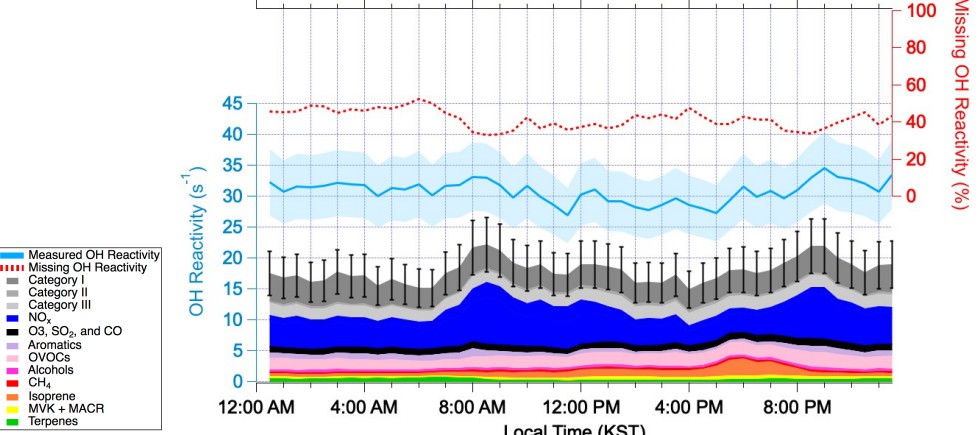

















Figure 3. Diurnal averages of OH reactivity during the stagnation period (A) from May 17th –
May 22nd in 2016 and the transport period (B) from 28 May – 1 June 2016. The measured and
calculated OH reactivity are on the left. The blue shading represents an uncertainty of 16.7% at
1σ. The black bars represent the propagated uncertainty of 20.1% at 1σ from calculated missing
OH reactivity. The percent missing OH reactivity is on the right axis.

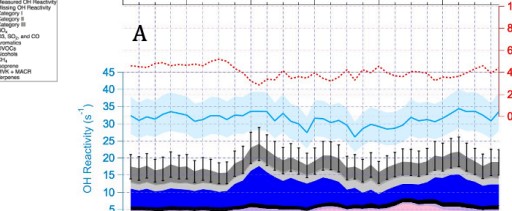 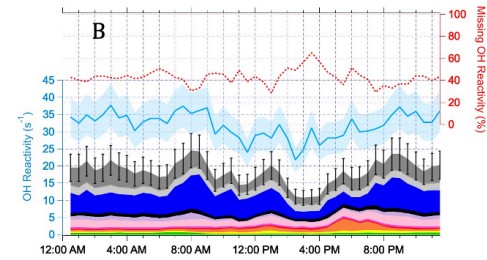


















Figure 4. The observed OH reactivity during the overall campaign, stagnation period, and
transport period.

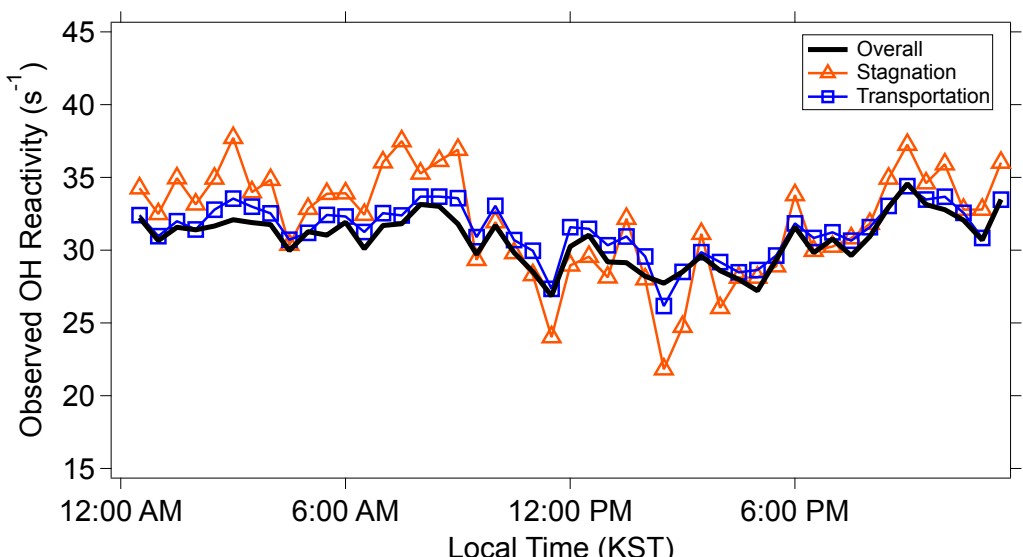














Figure 5. Diurnal profiles for different classes of trace gases during the different periods. A)
criteria pollutants $NO_x$, $O_3$, $SO_2$, and CO B) Aromatics, C) BVOCs, and D) OVOCs


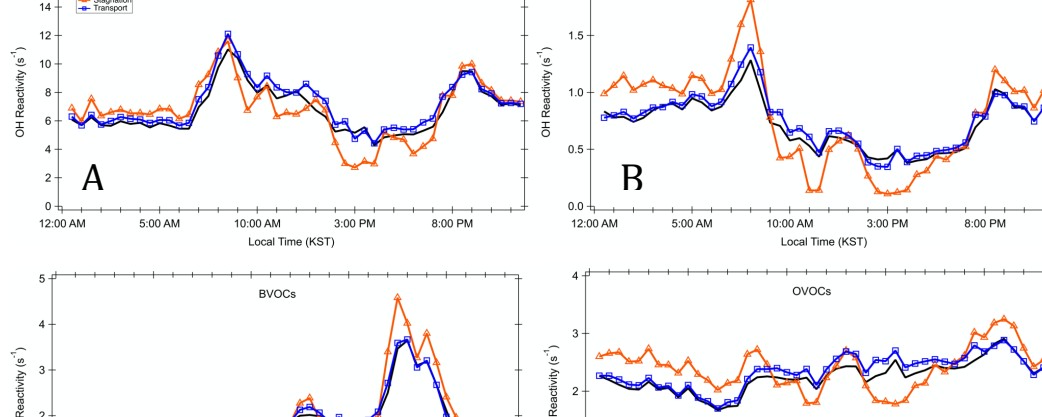
















Figure 6. Diurnal averages of the OH reactivity from the compounds in A) Category I, B)
Category II and C) Category III

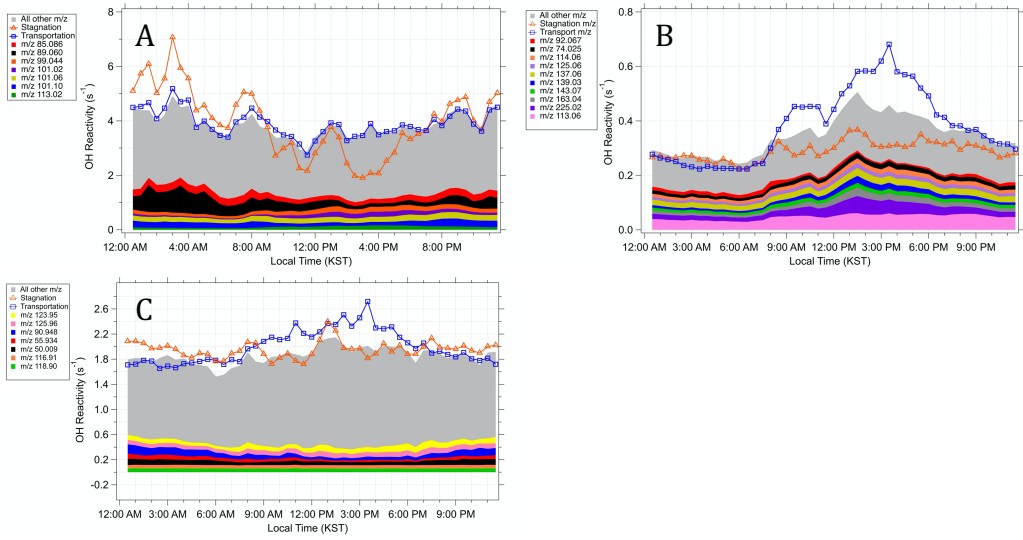
















Figure 7. The correlation between A) NOx OH reactivity and absolute missing OH reactivity and
B) percent missing OH reactivity


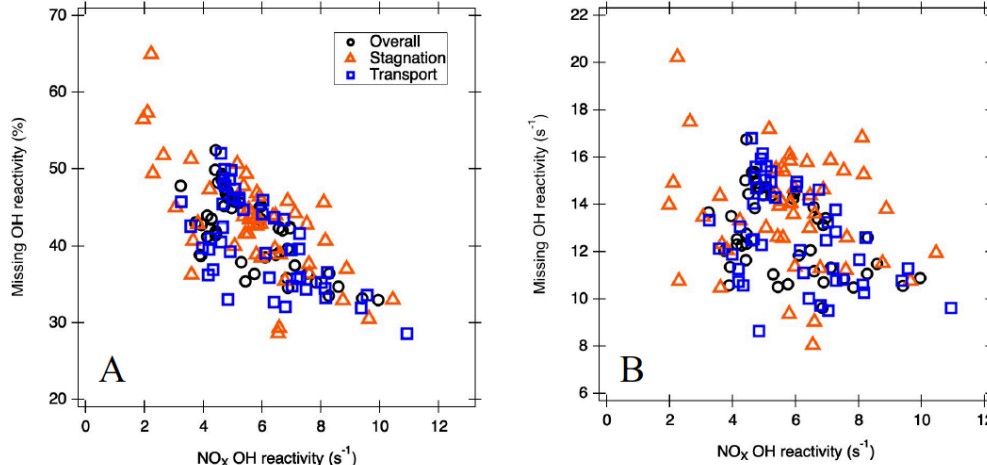
