# Peer review of "Contributions to OH reactivity from unexplored volatile organic compounds measured by PTR-ToF-MS– A case study in a suburban forest of the Seoul Metropolitan Area during KORUS-AQ 2016"

_Atmospheric Chemistry and Physics, 2020_

## Referee Comment (RC1) · Arnaud P. Praplan (Referee) · 26 Mar 2020

General comments:

The manuscript "Contributions to OH reactivity from unexplored volatile organic compounds measured by PTR-ToF-MS – A case study in a suburban forest of the Seoul Metropolitan Area during KORUS-AQ 2016" by Sanchez et al., present results from OH reactivity measured in spring 2016 in a Research Forest Station of Korean pine located

roughly 35km from the centre of Seoul. This site is influenced by both local biogenic emissions and transported anthropogenic emissions. OH reactivity measurement were operated with a CIMS-CRM instrument and Volatile Organic Compounds (VOCs) were monitored with a high resolution Proton-Transfer-Reaction Time-of-Flight Mass Spectrometer (PTR-ToF-MS). The authors found in addition to the most commonly monitored VOCs, 346 unidentified peaks that got assigned in three different categories in order to estimated their potential contribution to the observed missing OH reactivity. The missing reactivity is defined as the difference between measured total OH reactivity and the OH reactivity calculated from the known species quantified in the same air masses. While the addition of previously unexplored VOCs to the analysis decreased the amount of missing OH reactivity, still 42.0% of missing reactivity remained. The authors hypothesis is that this might be due to oxidation products from both biogenic and anthropogenic compounds.

I found the manuscript clear and concise. The introduction places this study in context in a comprehensive fashion. The methods and the results are mostly well explained and are sound. I found a few places, where the manuscript could benefit from further clarifications (see specific comments below), in particular explaining some assumptions taken for the calculations of OH reactivity from unidentified PTR-ToF-MS pseaks. Therefore, I would recommend ACP to publish this manuscript after minor corrections have been done.

Specific comments:

- p.3, ll.61-66: Are the authors aware of the more recent publications by Nölscher et al. (2012) and by Praplan et al. (2019) for additional OH reactivity measurements in Hyytiälä? Their findings were similar to the ones of Sinha et al. (2010) in terms of missing reactivity. Nölscher et al. found for instance that the highest missing reactivity values occurred during a period of higher temperature (labelled "stress" period). Praplan et al. also found among other things that including modelled oxidation compounds from the Master Chemical Mechanism only decreased the missing fraction by about

7% at most.

- p.8, ll.180-182: Fuchs et al. (2017) also found that some CRM instrument see an interference with additional O3 introduced from ambient air in the reactor. Has this interference also been investigated in the UCI CIMS-CRM? It would be a good thing to explicitly mention that this effect can be ruled out.

- p.9, ll.189-193: I understand that the detailed calibration procedures are detailed in another publication, but I believe that some additional information regarding the total OH reactivity instruments could be included. For instance there is no indication about what the pyrrole-to-OH ratio is during this campaign (and if it is the same as for the laboratory calibrations). Also from the Supplementary Material in Sanchez et al. (2018), if understand Fig. S1 correctly, the instrument response (without unit, but is it OH reactivity?) is about a quarter of what is expected and this is consistent over a wide spectrum of kOH values. I personally would find value in summarizing this in a couple of sentence in the present manuscript. Consequently, should we understand that the multi-point calibrations performed during the campaign were consistent with the measurements in Sanchez et al. (2018)?

- p.9, ll.202-203: Out of curiosity, is there a reason that the authors did attempt CIMS-CRM calibrations with the PTR-ToF-MS calibration mixture? (possibly before or after the measurement campaign, not necessarily during the campaign, as I understand that the instruments were physically located in two different shacks)

- p.11, ll.236-246: Category I: My main issue here is that, even if I am not familiar with the PTRwid software, I assume that it delivers "calculated molecular formula" (and not chemical structures) as mentioned by the authors. Looking at Table S2 of the Supplementary material I understand that for some of the molecular formula, there is only one sound structure possible. Could the authors discuss the chances that they found only one match in the NIST database, while other chemical structures, for which the database contain no data, could be possible? I also fail to understand how a

SAR-based estimate can be given for compound without any given "Possible ID". I understand that it is a lot of work to compile all these reaction rate coefficients, but the authors definitely ought to explain their assumptions better. Do authors rely on previous studies with the same instrument? Previous studies at the same site? I also could not find where the NIST value for k comes from for C4H4H+, for instance. Is it just in the wrong column? And why were the authors not able to use the SAR method for maleic anhydride? Is it a limitation of the method?

- p.11-12, ll.252-264: Category II: Should there be a regression adjusted to the m/z of isoprene for m/z 116.036 in Table S3, which is correlating with isoprene? (Also there is no mention of Table S3 and S4 in the main text)

- pp.12-13, ll. 278-281: Does this assumption rely on the fact that light hydrocarbons are not emitted locally? Can the authors be sure that there is no strong vertical gradient for these compounds regardless of the evolution of the boundary layer height?

Technical corrections:

p.27, Table 1, 3rd row: "Scientific" instead of "Scien8fic"

pp. 21-27: Check the format of the references. Sometimes "Doi" is used, sometimes "DOI", sometimes nothing; one link appears, "–" (l. 699) should probably be fixed, etc.

Supplementary material, Tables S3 and S4: I would use "peaks" instead of "compounds" in the tables' captions.

References:

Fuchs, H.; Novelli, A.; Rolletter, M.; Hofzumahaus, A.; Pfannerstill, E. Y.; Kessel, S.; Edtbauer, A.; Williams, J.; Michoud, V.; Dusanter, S.; Locoge, N.; Zannoni, N.; Gros, V.; Truong, F.; Sarda-Esteve, R.; Cryer, D. R.; Brumby, C. A.; Whalley, L. K.; Stone, D.; Seakins, P. W.; Heard, D. E.; Schoemaecker, C.; Blocquet, M.; Coudert, S.; Batut, S.; Fittschen, C.; Thames, A. B.; Brune, W. H.; Ernest, C.; Harder, H.; Muller, J. B. A.;

Elste, T.; Kubistin, D.; Andres, S.; Bohn, B.; Hohaus, T.; Holland, F.; Li, X.; Rohrer, F.; Kiendler-Scharr, A.; Tillmann, R.; Wegener, R.; Yu, Z.; Zou, Q. & Wahner, A. Comparison of OH reactivity measurements in the atmospheric simulation chamber SAPHIR Atmos. Meas. Tech., 2017, 10, 4023-4053.

Nölscher, A. C.; Yáñez-Serrano, A. M.; Wolff, S.; de Araujo, A. C.; Lavrič, J. V.; Kesselmeier, J. & Williams, J. Unexpected seasonality in quantity and composition of Amazon rainforest air reactivity Nature Comm., 2016, 7, 10383.

Praplan, A. P.; Tykkä, T.; Chen, D.; Boy, M.; Taipale, D.; Vakkari, V.; Zhou, P.; Petäjä, T. & Hellén, H. Long-term total OH reactivity measurements in a boreal forest Atmos. Chem. Phys., 2019, 19, 14431-14453.

Sanchez, D.; Jeong, D.; Seco, R.; Wrangham, I.; Park, J.-H.; Brune, W. H.; Koss, A.; Gilman, J.; de Gouw, J.; Misztal, P.; Goldstein, A.; Baumann, K.; Wennberg, P. O.; Keutsch, F. N.; Guenther, A. & Kim, S. Intercomparison of OH and OH reactivity measurements in a high isoprene and low NO environment during the Southern Oxidant and Aerosol Study (SOAS) Atmospheric Environment, 2018, 174, 227-236.

Sinha, V.; Williams, J.; Lelieveld, J.; Ruuskanen, T.; Kajos, M.; Patokoski, J.; Hellen, H.; Hakola, H.; Mogensen, D.; Boy, M.; Rinne, J. & Kulmala, M. OH Reactivity Measurements within a Boreal Forest: Evidence for Unknown Reactive Emissions Environ. Sci. Technol., 2010, 44, 6614-6620.

---

## Referee Comment (RC2) · Anonymous Referee #2 · 19 Apr 2020

This paper shows the results of the measurements of the total OH reactivity in the Tae-hwa Research Forest, southeast from the center of Seoul, South Korea. The authors indicate the existence of the missing OH sink and its sources. Due to the elucidation of the formation of the tropospheric ozone and SOA, this research is very important. In addition, this paper shows the detail results and discussions. Therefore, I recommend that this paper should be published if the minor revisions shows below is done.

«Comment» L.146-148: The mini tunable infrared laser direct absorption spectroscopy instrument (TILDAS) has applied to the measurements of HCHO, methane and so on. Is the TILDAS commercially available, or home-made? If the authors have already explain the detail of the TILDAS in some paper, the author should add a reference.

2.4. OH reactivity Calculation The authors grouped the 346 unidentified measured mass peaks into three categories. The authors should show the number of the uniden-tified peaks for each categories.

Figure 5(A) In general, anti-correlation between NO and O3 have been observed due to the reaction of NO with O3. In addition, ozone is formed secondary, which is similar with some OVOCs. Therefore, I recommend that the diurnal variation of the OH reactivity for O3 were distinguished with other pollutants.

(very) Minor: Font of the subscript of NO"x" is not unified. The authors should check it. In addition, the font of "k_OH" and "r^2" are not unified.

---

## Referee Comment (RC3) · Anonymous Referee #3 · 1 May 2020

This manuscript reports a comparison of OH reactivity measurements performed during the KORUS-AQ 2016 field campaign to OH reactivity values derived from the measurement of trace gases. In this study, the authors observed a large amount of missing OH reactivity, similar to that observed in previous campaigns focusing on areas impacted by biogenic emissions. The authors propose a procedure to assess the contribution of unidentified species detected by a high resolution PTRMS to the missing OH reactivity.

[Figure]

This reviewer thinks that this work is of interest for the scientific community and deserves publication. However, several aspects of this publication need to be straighten, especially the measurement quality of the CIMS-CRM instrument. I therefore recommend publication in ACP after the authors address the following comments:

Major comments:

L183-185: "This included a laboratory-built catalytic converter (Pt-wool at 350 °C) that minimized the interferences due to changes in air to prevent the interference from the difference in humidity for the zero air characterizations." – Heated Pt-wool will not remove NOx and may even lead to a higher NOx concentration at the output of the catalytic converter due to the oxidation of organic nitrate species present in ambient air. This has been reported in Hansen et al. (2015) where the authors used filters made of Purafil and activated charcoal upstream the catalytic converter to remove NOx. If NOx is not removed from the VOC-free air provided by the catalytic converter, the CRM measurements will be biased low. How did the authors remove NOx from the VOC-free air generated by their catalytic converter?

L186-189: "Additionally, interference from the OH recycling reaction of HO2 with NO in the glass chamber was prevented by removing OH reactivity data points that corresponded with ambient NO levels that exceeded 5 ppb to reflect results from a laboratory characterization study (Sanchez et al., 2018)." – The reviewer checked in Sanchez et al. (2018) but did not find the laboratory characterization mentioned in the above sentence. The authors should provide some information about the tests performed on the UCI CIMS-CRM instrument to characterize OH recycling. It is stated that OH reactivity measurements performed when NO was larger than 5 ppb were discarded. However, OH recycling does not stop below 5 ppb NO and still impacts the OH reactivity measurements. The impact will be instrument dependent and it is important to assess it on the UCI CIMS-CRM. Fuchs et al. (2017) report corrections of 10-20 s-1 at 10 ppb NO for 3 CRM instruments. Assuming that the correction scales with NO on this small range of concentrations, 5 ppb of NO would lead to a correction ranging from 5-10 s-1

for the instruments used in Fuchs et al., which is significant compared to the range of OH reactivity observed in the present study (20-50 s-1). In addition, NO2 was observed to generate measurement artifacts on some CRM instruments (Michoud et al., 2015; Fuchs et al. 2017), requiring corrections of 1.6-3 s-1 at a NO2 mixing ratio of 10 ppb. For the present study, Figure S2 reports mixing ratios ranging from 2-60 ppb for NO2. Was this issue investigated on the UCI CIMS-CRM?

L189-193: The authors refer the reader to Sanchez et al. (2018) for details on calibration procedures. The reviewer agree with the comments from the first reviewer that additional information on calibration procedures is needed in the present publication. What was the pyrrole-to-OH ratio in the CRM reactor during calibration experiments reported in Sanchez et al. (2018)? Why is there a significant intercept of 3.9 s-1 on the calibration curve (Sanchez et al., 2018)? What was the range of pyrrole-to-OH ratios during the field measurements reported in the present publication? Previous studies have shown that the CRM response can depend on the pyrrole-to-OH ratio (Michoud et al., 2015; Zannoni et al., 2015) and calibrations have to be performed over the range of pyrrole-to-OH ratios observed during field measurements. Can the authors comment on this aspect for the UCI CIMS-CRM?

L236-246: When a molecular formula was determined by PTRwid, how did the authors choose between the different possibilities for isomers? The reviewer checked how the OH rate constant would change between different isomers for C3H6O2 detected at the nominal m/z 75 (methyl acetate 3,5E-13 cm3/molecule/s; propanoic acid 1.2E-12; hydroxyacetone 3E-12; hydroxypropanal 3.5-5.5E-11; and others). The OH rate constant can span 2 orders of magnitude for these isomers. Could the authors comment about the choice made in Table S2 to use a kOH value of 1.2E-12 cm3/molecule/s for this molecular formula? This issue is even worse for ions detected at higher masses since more isomers have to be taken into account. What methodology did the authors used to select the rate constants reported in Table S2?

Minor comments:

L57-59: "Di Carlo et al. (2004) conducted a study in a mixed forest near Pellston, Michigan where they reported missing OH reactivity larger than observational uncertainty." – Please be quantitative. How much larger?

L115: The authors reviewed part of the OH reactivity literature for forested areas showing that variable agreements were observed between measured and calculated OH reactivity (from trace gas measurements or constrained O-D modeling), the disagreement being sometime attributed to (i) unknown emissions of VOC, (ii) oxidation products of primary VOC, or (iii) both. The reviewer would have liked to see a short discussion on the current limits in the identification of the missing OH reactivity. How can additional studies help improving our understanding of the missing OH coreactants? This will help motivating the present work.

L149-L151: "The OH reactivity and NOx analyzers were located in another nearby air-conditioned shack (3 m apart) and sampled air through an extended Teflon inlet line of 4 m ($\frac{1}{4}$" OD) from the ground with a flow rate of 4 sLpm resulting in a 0.5 second residence time." – Please indicate the height of the sampling inlet for the OH reactivity instrument and the NOx analyzers.

L161-162: The authors indicate a pyrrole+OH rate constant of 1.45E-10 cm3/molecule/s at 298 K. The rate constant recommended by Dillon et al. (2012) and Atkinson et al. (1984) is 1.2E-10 cm3/molecule/s at the same temperature. Why did the authors use a different rate constant?

L212-213: "we determined the VOC sensitivities using equation 1 (Eq 1)" – Equation 1 is not used to derive the VOC sensitivity but to calculate the concentration of uncalibrated species. Please rephrase.

L219: Equation 1 is confusing. Is "ncps(benzene)/ppb" the normalized signal (corrected for mass discrimination) generated by 1 ppb of benzene? What is the factor of 11.94? Please clarify this equation. How was the ion transmission curve characterized on the PTRMS to correct the normalized VOC signals for mass discrimination?

L247-251: Please indicate the range of R2 factors for the correlations. Also indicate R2 factors in Table S3.

L253-255: "The compounds were grouped into 5 m/z bins and the average kOH of each bin was calculated. The green triangles represent 5 m/z binned averages from these compounds plotted with their respective average kOH." – The reviewer does not understand what was done here. Please clarify this section. Also provide additional information on what is shown in Figure S2. What is the purpose of the different regression lines?

L371-374: The authors are discussing the potential species detected at m/z 83.085. This mass is not shown in Figure 6 and the authors may want to add it.

L417-420: "As NOx illustrates a conspicuous temporal variation that appears to correlate with the fraction of missing OH reactivity, while observed OH reactivity and calculated OH reactivity from VOCs indicate a less pronounced diurnal difference. " - The reviewer does not understand what is meant here. Please clarify/rephrase this sentence.

L428-430: "Moreover, due to the different inlet configurations for OH reactivity and VOC observations, their contributions towards observed and calculated OH reactivity may not have been consistently evaluated." – This sentence also need to be clarified. Are the authors discussing the impact of the inlet on the measured trace gases and OH reactivity?

Table 1: Please include detection limit and time resolution for each instrument. Brand and model of PTRMS?

Figure 2: How did the authors assess the uncertainty associated to the calculated OH reactivity? What were the sources of errors factored in the calculations? What uncertainty did the authors consider on the OH reactivity from categories I-III?

Figure 3: The authors indicate an uncertainty of 20.1% for the missing OH reactivity?

[Figure]

Since the uncertainty stated for the OH reactivity measurements is 16.7%, a quadratic propagation of errors allows calculating that the authors considered an uncertainty of approximately 11% (1 sigma) on the calculated OH reactivity. This seems a bit low since this uncertainty should account for errors associated to measured trace gases and tabulated reaction rate constants, the latter being already in the range 10-25% at 1 sigma. For the measured trace gases, it is stated that the error on NOx, which account for a large fraction of the OH reactivity, is 20% at 1 sigma. While the uncertainty associated to PTRMS measurements of calibrated compounds is within 5-10%, the uncertainty on mixing ratios derived from masses where a default proton transfer rate constant of 3E-9 cm3/s was used will not be better than 25% (1 sigma). How was the uncertainty on the calculated OH reactivity derived?

Edits:

L74 : Âń . . . alkanes, alkenes and other observed trace gases such as. . ." should read "alkanes and alkenes and other observed trace gases such as. . ."

L358: "m/z 101" – Should it read "m/z 101.10".

L 458: "Further study is required ..." should read "Further studies are required . . ."

References:

Atkinson, R., Aschmann, S. M., Winer, A. M. and Carter, W. P. L.: Rate constants for the gas phase reactions of OH radicals and O3 with pyrrole at 295 ± 1 K and atmospheric pressure, Atmos. Environ. 1967, 18, 2105–2107, doi:10.1016/0004-6981(84)90196-3, 1984.

Dillon, T., Tucceri, M., Dulitz, K., Horowitz, A., Vereecken, L., and Crowley, J.: Reaction of Hydroxyl Radicals with C4H5N (Pyrrole): Temperature and Pressure Dependent Rate Coefficients, J. Phys. Chem. A, 116, 6051–6058, doi:10.1021/jp211241x, 2012.

Fuchs, H., Novelli, A., Rolletter, M., Hofzumahaus, A., Pfannerstill, E. Y., Kessel, S., Edtbauer, A., Williams, J., Michoud, V., Dusanter, S., Locoge, N., Zannoni, N., Gros,

V., Truong, F., Sarda-Esteve, R., Cryer, D. R., Brumby, C. A., Whalley, L. K., Stone, D., Seakins, P. W., Heard, D. E., Schoemaecker, C., Blocquet, M., Coudert, S., Batut, S., Fittschen, C., Thames, A. B., Brune, W. H., Ernest, C., Harder, H., Muller, J. B. A., Elste, T., Kubistin, D., Andres, S., Bohn, B., Hohaus, T., Holland, F., Li, X., Rohrer, F., Kiendler-Scharr, A., Tillmann, R., Wegener, R., Yu, Z., Zou, Q., and Wahner, A.: Comparison of OH reactivity measurements in the atmospheric simulation chamber SAPHIR, Atmos. Meas. Tech., 10, 4023–4053, https://doi.org/10.5194/amt-10-4023-2017, 2017.

Hansen, R. F., Blocquet, M., Schoemaecker, C., Léonardis, T., Locoge, N., Fittschen, C., Hanoune, B., Stevens, P. S., Sinha, V., and Dusanter, S.: Intercomparison of the comparative reactivity method (CRM) and pump–probe technique for measuring total OH reactivity in an urban environment, Atmos. Meas. Tech., 8, 4243–4264, https://doi.org/10.5194/amt-8-4243-2015, 2015.

Michoud, V., Hansen, R. F., Locoge, N., Stevens, P. S., and Dusanter, S.: Detailed characterizations of the new Mines Douai comparative reactivity method instrument via laboratory experiments and modeling, Atmos. Meas. Tech., 8, 3537–3553, https://doi.org/10.5194/amt-8-3537-2015, 2015.

Zannoni, N., Dusanter, S., Gros, V., Sarda Esteve, R., Michoud, V., Sinha, V., Locoge, N., and Bonsang, B.: Intercomparison of two comparative reactivity method instruments inf the Mediterranean basin during summer 2013, Atmos. Meas. Tech., 8, 3851–3865, https://doi.org/10.5194/amt-8-3851-2015, 2015.

---

## Author Comment (AC1) · 27 Jul 2020

Please see the attached..

Please also note the supplement to this comment:
https://acp.copernicus.org/preprints/acp-2020-174/acp-2020-174-AC1-supplement.pdf

---

## Author Response (AR1)

We appreciate constructive comments from Dr. Praplan and two other anonymous referees. We summarized our responses to each comment as below. We have specified how we have reflected the reviewers' comments by indicating line numbers in the track-change version of the revision.

Reviewer 1.

The manuscript "Contributions to OH reactivity from unexplored volatile organic com- pounds measured by PTR-ToF-MS – A case study in a suburban forest of the Seoul Metropolitan Area during KORUS-AQ 2016" by Sanchez et al., present results from OH reactivity measured in spring 2016 in a Research Forest Station of Korean pine located roughly 35km from the centre of Seoul. This site is influenced by both local biogenic emissions and transported anthropogenic emissions. OH reactivity measurement were operated with a CIMS-CRM instrument and Volatile Organic Compounds (VOCs) were monitored with a high resolution Proton-Transfer-Reaction Time-of-Flight Mass Spectrometer (PTR-ToF-MS). The authors found in addition to the most commonly monitored VOCs, 346 unidentified peaks that got assigned in three different categories in order to estimated their potential contribution to the observed missing OH reactivity. The missing reactivity is defined as the difference between measured total OH reactivity and the OH reactivity calculated from the known species quantified in the same air masses. While the addition of previously unexplored VOCs to the analysis decreased the amount of missing OH reactivity, still 42.0% of missing reactivity remained. The authors hypothesis is that this might be due to oxidation products from both biogenic and anthropogenic compounds.
I found the manuscript clear and concise. The introduction places this study in context in a comprehensive fashion. The methods and the results are mostly well explained and are sound. I found a few places, where the manuscript could benefit from further clarifications (see specific comments below), in particular explaining some assumptions taken for the calculations of OH reactivity from unidentified PTR-ToF-MS peaks. Therefore, I would recommend ACP to publish this manuscript after minor corrections have been done.
Specific comments:

- p.3, ll.61-66: Are the authors aware of the more recent publications by Nölscher et al. (2012) and by Praplan et al. (2019) for additional OH reactivity measurements in Hyytiälä? Their findings were similar to the ones of Sinha et al. (2010) in terms of missing reactivity. Nölscher et al. found for instance that the highest missing reactivity values occurred during a period of higher temperature (labelled "stress" period). Praplan et al. also found among other things that including modelled oxidation compounds from the Master Chemical Mechanism only decreased the missing fraction by about7% at most.

In the revision we have included the suggested references. Now it is read:

Nolscher et al. (2012) observed the highest level of missing OH reactivity during a heat wave episode, possibly inducing a stress emission response from the local forest. A comprehensive analysis by Praplan et al. (2019) using a long-term observation dataset and a photochemical model framework with the Master Chemical Mechanism illustrates that the model simulated oxidation compound contribution can only contribute 7 % of missing OH reactivity.  (line 71 - 78 in the track change version)

- p.8, ll.180-182: Fuchs et al. (2017) also found that some CRM instrument see an interference with additional O3 introduced from ambient air in the reactor. Has this interference also been investigated in the UCI CIMS-CRM? It would be a good thing to explicitly mention that this effect can be ruled out.

We conduct one experiment by adding standard (propene) when ozone was 60 pppb and 120 ppb during MAPS-Seoul campaign and does not see the statistical difference but we did not follow up more systematic fashion as Fuchs et al. did. This is something we should address in the up-coming lab study. We have included that this discussion in the revised manuscript. Now it is read:

In addition, Fuchs et al. (2017) also described a potential interference from ambient $O_3$ in some CRM systems. In the 2015 field campaigns conducted in Seoul South Korea (Kim et al., 2016), we conducted a standard addition experiment for the propene standard for additional ~ 30 s$^{-1}$ in two different ozone environment 65 ppb and 123 ppb. The outcome illustrates an agreement between two additions within the analytical uncertainty. (line 263 - 267 in the track change version)

- p.9, ll.189-193: I understand that the detailed calibration procedures are detailed in another publication, but I believe that some additional information regarding the total OH reactivity instruments could be included. For instance there is no indication about what the pyrrole-to-OH ratio is during this campaign (and if it is the same as for the laboratory calibrations). Also from the Supplementary Material in Sanchez et al. (2018), if understand Fig. S1 correctly, the instrument response (without unit, but is it OH reactivity?) is about a quarter of what is expected and this is consistent over a wide spectrum of kOH values. I personally would find value in summarizing this in a couple of sentence in the present manuscript. Consequently, should we understand that the multi-point calibrations performed during the campaign were consistent with the measurements in Sanchez et al. (2018)?

The Y-axis in the calibration curve, presented in Sanchez et al. (2018), used an arbitrary unit as we were using pyrrole to OH ration of 3:1, which would not warrant the pseudo first order reaction regime. Since the the publication of Sanchez et al. (2018), we have maintained identical configuration and sensitivity has been maintained within 30 %. This description has been included in the revised manuscript.

Now it is read

We consistently kept the pyrrole to OH ratio at 3:1 and so did not achieve a pseudo first order relationship.(line 214 - 215 in the track change version)

- p.9, ll.202-203: Out of curiosity, is there a reason that the authors did attempt CIMS-CRM calibrations with the PTR-ToF-MS calibration mixture? (possibly before or after the measurement campaign, not necessarily during the campaign, as I understand that the instruments were physically located in two different shacks)

This calibration standard mixture is only used for PTR-ToF-MS calibration. We have make it clear in the revised manuscript. We did not use the multi-component mixture for the CIMS-CRM calibration.

Now it is read

. This standard mixture was only used for the PTR-ToF-MS calibration and not the CRM-CIMS calibration. (line 279 -280 in the track change version)

- p.11, ll.236-246: Category I: My main issue here is that, even if I am not familiar with the PTRwid software, I assume that it delivers "calculated molecular formula" (and not chemical structures) as mentioned by the authors. Looking at Table S2 of the Supplementary material I understand that for some of the molecular formula, there is only one sound structure

possible. Could the authors discuss the chances that they found only one match in the NIST database, while other chemical structures, for which the database contain no data, could be possible? I also fail to understand how a SAR-based estimate can be given for compound without any given "Possible ID". I understand that it is a lot of work to compile all these reaction rate coefficients, but the authors definitely ought to explain their assumptions better. Do authors rely on previous studies with the same instrument? Previous studies at the same site? I also could not find where the NIST value for k comes from for C4H4H+, for instance. Is it just in the wrong column? And why were the authors not able to use the SAR method for maleic anhydride? Is it a limitation of the method?

Most of cases, we encountered more than one possible structures. Then we go over the previous publications such as Williams et al. (2001), de Gouw et al. (2003), Jordan et al. (2009), Ruuskanen et al. (2011) Müller et al. (2012), and Koss et al. (2017) to list all the possible compounds, reported in the ambient air before. Then we took a median for the rate constant. The detailed list and process is well described in Sanchez (2019). We have included this discussion in the revised manuscript. For C4H4H+ we categorize it as vinylacetylene (C4H4) with a reported rate constant. We also have included this in the revised manuscript. For that specific case of m/z 75, the reviewer raised, we found that there are five possibilities - hydroxyacetone, propionic acid, unidentified oxidation product fragments of terpenes, methyl acetate, and ethyl formate. Among them, we choose the median reaction constant. The detailed selection process for all masses is thoroughly discussed Dianne Sanchez's Ph.D. dissertation. The embargo period has been over since July 8th of 2020 so we add this to the reference and have included detailed discussion on this.

Now it is read

As the only information we have is the molecular composition, we identified multiple isomers with different functional groups and thus different reactivity. We have extensively reviewed previous publications (Williams et al., 2001;De Gouw et al., 2003;de Gouw and Warneke, 2007a;Jordan et al., 2009a;Ruuskanen et al., 2011;Muller et al., 2012;Koss et al., 2017a) identifying ambient VOCs using PTR-MS with both quadrupole and time-of-flight systems to identify possible compounds. For example, for the m/z of 75.043, there are four possible compounds including hydroxy acetone, propionic acid, methyl acetate, and ethyl formate. We used the median reaction constant for the set of possible compounds. The detailed description of the OH reaction constant determination process for the Category I peaks is described in Sanchez (2019) (line 329 -338 in the track change version).

- p.11-12, ll.252-264: Category II: Should there be a regression adjusted to the m/z of isoprene for m/z 116.036 in Table S3, which is correlating with isoprene? (Also there is no mention of Table S3 and S4 in the main text)

We have added the regression line in Figure S1 for the $k_{OH}$ calculation of the isoprene type compounds in the revised manuscript.

- pp.12-13, ll. 278-281: Does this assumption rely on the fact that light hydrocarbons are not emitted locally? Can the authors be sure that there is no strong vertical gradient for these compounds regardless of the evolution of the boundary layer height?

We assumed the homogeneous distribution of small alkanes and alkenes in the boundary layer because they have relatively long chemical lifetime in comparison with the mixing time scale of the convective boundary layer. As the referee suggested, certainly, no

known strong emission sources of those compounds is the important factor. We have included this discussion in the revised manuscript.

Not it is read

In this suburban forest, we do not think there is any substantial emission sources of these relatively long-lived VOCs. (line 411-412 in the track change version)

Technical corrections:
p.27, Table 1, 3rd row: "Scientific" instead of "Scien8fic"

We have corrected this in the revised manuscript.

pp. 21-27: Check the format of the references. Sometimes "Doi" is used, sometimes "DOI", sometimes nothing; one link appears, "–" (l. 699) should probably be fixed, etc. Supplementary material, Tables S3 and S4: I would use "peaks" instead of "compounds" in the tables' captions.

We have corrected this in the revised manuscript.

Reviewer 2.

This paper shows the results of the measurements of the total OH reactivity in the Taehwa Research Forest, southeast from the center of Seoul, South Korea. The authors indicate the existence of the missing OH sink and its sources. Due to the elucidation of the formation of the tropospheric ozone and SOA, this research is very important. In addition, this paper shows the detail results and discussions. Therefore, I recommend that this paper should be published if the minor revisions shows below is done.

«Comment» L.146-148: The mini tunable infrared laser direct absorption spectroscopy instrument (TILDAS) has applied to the measurements of HCHO, methane and so on. Is the TILDAS commercially available, or home-made? If the authors have already explain the detail of the TILDAS in some paper, the author should add a reference.

We added a reference and manufacturer information about the instrument (Table 1).

2.4. OH reactivity Calculation The authors grouped the 346 unidentified measured mass peaks into three categories. The authors should show the number of the unidentified peaks for each categories.

We have included the information about the number of peaks for each category (Line number 327, 353, and 368 in the track change version).

Figure 5(A) In general, anti-correlation between NO and O3 have been observed due to the reaction of NO with O3. In addition, ozone is formed secondary, which is similar with some OVOCs. Therefore, I recommend that the diurnal variation of the OH reactivity for O3 were distinguished with other pollutants.

As shown in Figure 2 and Figure 3, the OH reactivity contribution of ozone is very minor. Most of the OH reactivity contribution in this category comes from NOX (mostly NO2). We clarify this point in the revised manuscript.

Now it is read

Furthermore, no significant variance of the different classes of reactive gases such as criteria air pollutants (CO, $NO_x$, $O_3$, and $SO_2$), mostly contributed by $NO_X$, OVOCs (acetone, acetaldehyde, formaldehyde, methylglyoxal, methanol, methyl ethyl ketone), aromatics (benzene, toluene, xylenes, styrene, benzaldehyde, trimethylbenzenes), and BVOCs (isoprene, monoterpenes, sesquiterpenes, MVK+MACR) was observed during the different periods (Figure 5).  (Line number 460 - 464 in the track change version)

(very) Minor: Font of the subscript of NO"x" is not unified. The authors should check it. In addition, the font of "k_OH" and "r^2" are not unified.

We double-checked throughout the manuscript to make the notation consistent in the revised manuscript.

Reviewer 3.
This manuscript reports a comparison of OH reactivity measurements performed dur- ing the KORUS-AQ 2016 field campaign to OH reactivity values derived from the mea- surement of trace gases. In this study, the authors observed a large amount of missing OH reactivity, similar to that observed in previous campaigns focusing on areas impacted by biogenic emissions. The authors propose a procedure to assess the contribution of unidentified species detected by a high resolution PTRMS to the missing OH reactivity. This reviewer thinks that this work is of interest for the scientific community and de- serves publication. However, several aspects of this publication need to be straighten, especially the measurement quality of the CIMS-CRM instrument. I therefore recommend publication in ACP after the authors address the following comments:
Major comments:

L183-185: "This included a laboratory-built catalytic converter (Pt-wool at 350 ○C) that minimized the interferences due to changes in air to prevent the interference from the difference in humidity for the zero air characterizations." – Heated Pt-wool will not remove NOx and may even lead to a higher NOx concentration at the output of the catalytic converter due to the oxidation of organic nitrate species present in ambient air. This has been reported in Hansen et al. (2015) where the authors used filters made of Purafil and activated charcoal upstream the catalytic converter to remove NOx. If NOx is not removed from the VOC-free air provided by the catalytic converter, the CRM measurements will be biased low. How did the authors remove NOx from the VOC-free air generated by their catalytic converter?

We have included more details on our approach to handle NOX. Basically, we did a laboratory calibration experiments with different NOX. Our goal was to discern a threshold level that disrupts the calibration curve by increasing NO concentration. The threshold was empirically determined as 5 ppb. In this level, the slope of the calibration curve is statistically deviated from the slope that is observed in the standard matrix with 5 ppb or lower NO levels. We did not add any scrubber except a catalytic converter.

Now it is read:

Hansen et al. (2015) illustrated that $NO_X$ may be generated from the catalytic converter. To prevent potential $NO_X$ interferences, they used a scrubber with Purafil and activated

charcoal, which will modulate the humidity in the sample. Our approach to this type of interference has been to determine the maximum NO level, noticeably interfering with the calibration regression line shown in Sanchez et al. (2018). Laboratory tests indicate that the statistical agreement started to veer off when the NO level is 5 ppb in 1 of the linear regression. In addition, Kim et al. (2016) achieved an OH reactivity budget closure in high $NO_2$ condition, which implies no significant interferences from $NO_2$. However, in response to the Fuchs et al. (2017) observation that various CRM configurations suffer from different levels of $NO_X$ interferences, we plan to conduct more systematic $NO_X$ interference tests to more accurately characterize this system. (Line 204-213 in the track change version)

L186-189: "Additionally, interference from the OH recycling reaction of HO2 with NO in the glass chamber was prevented by removing OH reactivity data points that corre- sponded with ambient NO levels that exceeded 5 ppb to reflect results from a laboratory characterization study (Sanchez et al., 2018)." – The reviewer checked in Sanchez et al. (2018) but did not find the laboratory characterization mentioned in the above sentence. The authors should provide some information about the tests performed on the UCI CIMS-CRM instrument to characterize OH recycling. It is stated that OH reactivity measurements performed when NO was larger than 5 ppb were discarded. However, OH recycling does not stop below 5 ppb NO and still impacts the OH reactivity mea- surements. The impact will be instrument dependent and it is important to assess it on the UCI CIMS-CRM. Fuchs et al. (2017) report corrections of 10-20 s-1 at 10 ppb NO for 3 CRM instruments. Assuming that the correction scales with NO on this small range of concentrations, 5 ppb of NO would lead to a correction ranging from 5-10 s-1 for the instruments used in Fuchs et al., which is significant compared to the range of OH reactivity observed in the present study (20-50 s-1). In addition, NO2 was observed to generate measurement artifacts on some CRM instruments (Michoud et al., 2015; Fuchs et al. 2017), requiring corrections of 1.6-3 s-1 at a NO2 mixing ratio of 10 ppb. For the present study, Figure S2 reports mixing ratios ranging from 2-60 ppb for NO2. Was this issue investigated on the UCI CIMS-CRM?

As described above, the upper limit of NO was empirically determined by a series of laboratory calibration as described above. We did not conduct a laboratory test for NO2 interference. However, our observations in a high NO2 environment (Seoul, Kim et al. (2016) Faraday Discussion) illustrate no indication of significant interference from NO2. We have included the detailed discussion in the revised manuscript (204-213 as shown above response).

L189-193: The authors refer the reader to Sanchez et al. (2018) for details on calibration procedures. The reviewer agree with the comments from the first reviewer that additional information on calibration procedures is needed in the present publication. What was the pyrrole-to-OH ratio in the CRM reactor during calibration experiments reported in Sanchez et al. (2018)? Why is there a significant intercept of 3.9 s-1 on the calibration curve (Sanchez et al., 2018)? What was the range of pyrrole-to-OH ratios during the field measurements reported in the present publication? Previous studies have shown that the CRM response can depend on the pyrrole-to-OH ratio (Michoud et al., 2015; Zannoni et al., 2015) and calibrations have to be performed over the range of pyrrole-to-OH ratios observed during field measurements. Can the authors comment on this aspect for the UCI CIMS-CRM?

For the response of Dr. Praplan's comment, we have added detailed information about the calibration procedures in the revised manuscript. The intercept is from the zero air mixture. We have used a compressed air from the building with a charcoal scrubber, which might not

effectively remove all the reactive compounds in the house compressed air supply. We have also included a detailed description on pyrrole to OH ratios and how we maintained the ratios over time.

Now it is read:

      We consistently kept the pyrrole to OH ratio at 3:1 and so did not achieve  a pseudo first order relationship. (Line 214 - 215 in the track change version)

L236-246: When a molecular formula was determined by PTRwid, how did the authors choose between the different possibilities for isomers? The reviewer checked how the OH rate constant would change between different isomers for C3H6O2 detected at the nominal m/z 75 (methyl acetate 3,5E-13 cm3/molecule/s; propanoic acid 1.2E-12; hydroxyacetone 3E-12; hydroxypropanal 3.5-5.5E-11; and others). The OH rate constant can span 2 orders of magnitude for these isomers. Could the authors comment about the choice made in Table S2 to use a kOH value of 1.2E-12 cm3/molecule/s for this molecular formula? This issue is even worse for ions detected at higher masses since more isomers have to be taken into account. What methodology did the authors used to select the rate constants reported in Table S2? Minor comments:

      As we responded to Dr. Praplan, we have extensively searched previous publications reporting different m/z detected in laboratory and field experiments. For that specific case of m/z 75, the reviewer raised, we found that there are five possibilities - hydroxyacetone, propionic acid, unidentified oxidation product fragments of terpenes, methyl acetate, and ethyl formate. Among them, we choose the median reaction constant. The detailed selection process for all masses is thoroughly discussed Dianne Sanchez's Ph.D. dissertation. The embargo period will be over by July 8th of 2020 so we add this to the reference and have included detailed discussion on this.

Now it is read:

      As the only information we have is the molecular composition, we identified multiple isomers with different functional groups and thus different reactivity. We have extensively reviewed previous publications (Williams et al., 2001;De Gouw et al., 2003;de Gouw and Warneke, 2007a;Jordan et al., 2009a;Ruuskanen et al., 2011;Muller et al., 2012;Koss et al., 2017a) identifying ambient VOCs using PTR-MS with both quadrupole and time-of-flight systems to identify possible compounds. For example, for the m/z of 75.043, there are four possible compounds including hydroxy acetone, propionic acid, methyl acetate, and ethyl formate. We used the median reaction constant for the set of possible compounds. The detailed description of the OH reaction constant determination process for the Category I peaks is described in Sanchez (2019). (Line 329 -338 in the track change version)

L57-59: "Di Carlo et al. (2004) conducted a study in a mixed forest near Pellston, Michigan where they reported missing OH reactivity larger than observational uncertainty." – Please be quantitative. How much larger?

      We have included quantitative discussion about the uncertainty.

      Now it is read:

Di Carlo et al. (2004) conducted a study in a mixed forest near Pellston, Michigan where they reported missing OH reactivity (~ 30 %) larger than observational uncertainty. (Line 61 - 63 in the track change version.)

L115: The authors reviewed part of the OH reactivity literature for forested areas showing that variable agreements were observed between measured and calculated OH reactivity (from trace gas measurements or constrained O-D modeling), the disagreement being sometime attributed to (i) unknown emissions of VOC, (ii) oxidation products of primary VOC, or (iii) both. The reviewer would have liked to see a short discussion on the current limits in the identification of the missing OH reactivity. How can additional studies help improving our understanding of the missing OH coreactants? This will help motivating the present work.

We added a sentence read as "This *status quo* urges us to take a convergent approach by effectively integrating observational results from novel instrumentations and model outcomes." In the revised manuscript ( the track change version line 126 - 128).

L149-L151: "The OH reactivity and NOx analyzers were located in another nearby air-conditioned shack (3 m apart) and sampled air through an extended Teflon inlet line of 4 m (14" OD) from the ground with a flow rate of 4 sLpm resulting in a 0.5 second residence time." – Please indicate the height of the sampling inlet for the OH reactivity instrument and the NOx analyzers.

We have included the information (3.5 m AGL) in the revised manuscript (track change version line166-167).

L161-162: The authors indicate a pyrrole+OH rate constant of 1.45E-10 cm3/molecule/s at 298 K. The rate constant recommended by Dillon et al. (2012) and Atkinson et al. (1984) is 1.2E-10 cm3/molecule/s at the same temperature. Why did the authors use a different rate constant?

We have corrected the rate constant accordingly in the revised manuscript (178-179).

Not it is read

The CRM method measures total OH reactivity by quantifying the relative loss of pyrrole, a highly reactive gas ($k_{OH+ pyrrole}$ = 1.07 × 10$^{-10}$ cm$^3$ molecule$^{-1}$ s$^{-1}$ at 298 K (Dillon et al., 2012)) that is rarely found in the atmosphere (Sinha et al., 2008b). (Line 178 - 179 in the track change version)

L212-213: "we determined the VOC sensitivities using equation 1 (Eq 1)" – Equation 1 is not used to derive the VOC sensitivity but to calculate the concentration of uncalibrated species. Please rephrase.

We actually used the relationship not necessarily used the equation for the mixing ratio assessment. We have revised the manuscript to minimize any confusion (296-301).

The application of this equation can be justified since PTRwid provides the mass discrimination corrected counts as a final computational product.

$$ppb_{VOC} = ncps_{VOC} \times k_{benzene}/k_{VOC} \times 1/11.94 \; ncps \; ppb^{-1} \; Eq1$$
where, 11.94 ncps ppb$^{-1}$ is the assessed sensitivity of benzene.

(line 296 - 301 in the track change version)

L219: Equation 1 is confusing. Is "ncps(benzene)/ppb" the normalized signal (corrected for mass discrimination) generated by 1 ppb of benzene? What is the factor of 11.94? Please clarify this equation. How was the ion transmission curve characterized on the PTRMS to correct the normalized VOC signals for mass discrimination?

Mass discrimination has been corrected while we process the raw data as a part of the data mining process. We have clarified the point and the equation in the revised manuscript as shown in the previous response.

L247-251: Please indicate the range of R2 factors for the correlations. Also indicate R2 factors in Table S3.

We have included the information in the revised manuscript (Table S3 in the revised manuscript).

Now it is read

Nonetheless, this group of compounds illustrated a positive correlation ($R^2$ = 0.19 to 0.88) with either anthropogenic (benzene, toluene) or biogenic (MVK+MACR and monoterpenes) VOCs.

L253-255: "The compounds were grouped into 5 m/z bins and the average kOH of each bin was calculated. The green triangles represent 5 m/z binned averages from these compounds plotted with their respective average kOH." – The reviewer does not understand what was done here. Please clarify this section. Also provide additional information on what is shown in Figure S2. What is the purpose of the different regression lines?

We have further described in this section for clarifications in the revised manuscript.

Now it is read:

More specifically, we assume that $k_{OH}$ is linearly correlated with m/z. To apply this linear relationship, the compounds with known $k_{OH}$ were grouped into 5 *m/z* bins and the average $k_{OH}$ of each bin was calculated. (Line 359 - 362 in the track change version)

L371-374: The authors are discussing the potential species detected at m/z 83.085. This mass is not shown in Figure 6 and the authors may want to add it.

We have included the trace in the revised manuscript (Figure 6 in the revised manuscript).

L417-420: "As NOx illustrates a conspicuous temporal variation that appears to correlate with the fraction of missing OH reactivity, while observed OH reactivity and calculated OH reactivity from VOCs indicate a less pronounced diurnal difference. " - The reviewer does not understand what is meant here. Please clarify/rephrase this sentence.

We have clarify it in the revised manuscript.

Now it is read:

This leads us to speculate that there is a consistent presence of unquantified trace gases, likely oxidation products of both anthropogenic and biogenic VOCs as we explored the origin of the unexplored peaks causing missing OH reactivity. In other words, $NO_x$ is relatively well measured with a highly pronounced temporal variation that determines the percentage of missing OH reactivity. (line 550 - 553 in the track change version)

L428-430: "Moreover, due to the different inlet configurations for OH reactivity and VOC observations, their contributions towards observed and calculated OH reactivity may not have been consistently evaluated." – This sentence also need to be clarified. Are the authors discussing the impact of the inlet on the measured trace gases and OH reactivity?

We have added a relevant reference in the revised manuscript.

Now it is read:

Moreover, due to the different inlet configurations for OH reactivity and VOC observations, their contributions towards observed and calculated OH reactivity may not have been consistently evaluated (e.g. Sanchez et al. (2018)). (line 561 - 563 in the track change version)

Table 1: Please include detection limit and time resolution for each instrument. Brand and model of PTRMS?

We have included the information in the revised manuscript (Table 1 in the revised manuscript).

Figure 2: How did the authors assess the uncertainty associated to the calculated OH reactivity? What were the sources of errors factored in the calculations? What uncertainty did the authors consider on the OH reactivity from categories I-III?

We have included a discussion about uncertainty associated with calculated OH reactivity in the revised manuscript.

Now it reads

There are two components that need to be considered for the assessment of uncertainty associated with calculated OH reactivity: the concentration and the reaction constants with OH. The uncertainty of the observed trace gases is in the range of 5 % to 20 % as shown in Table 1 and is associated with the rate constants from laboratory experiments (Atkinson et al., 2006). Combining 15 % uncertainty from reaction constants and 13.5 % from trace gas observations results in 20 % of uncertainty in calculated OH reactivity. This should be considered as a conservative estimate as most VOC concentrations and associated rate constants are empirically estimated. (line 376 - 383 in the track change manuscript)

Figure 3: The authors indicate an uncertainty of 20.1% for the missing OH reactivity? Since the uncertainty stated for the OH reactivity measurements is 16.7%, a quadratic propagation of errors allows calculating that the authors considered an uncertainty of approximately 11% (1 sigma) on the calculated OH reactivity. This seems a bit low since this uncertainty should account for errors associated to measured trace gases and tabulated reaction rate constants, the latter being already in the range 10-25% at 1 sigma. For the measured trace gases, it is stated that the error on NOx, which account for a large fraction of the OH reactivity, is 20% at 1 sigma. While the uncertainty associated to PTRMS measurements of calibrated compounds is within 5-10%, the uncertainty on mixing ratios derived from masses where a default proton transfer rate constant of 3E-9 cm3/s was used will not be better than 25% (1 sigma). How was the uncertainty on the calculated OH reactivity derived?

We acknowledge that it was not entirely clear about the error bars in the figure. We have clarified the confusion by revising the figure caption as

Diurnal averages of OH reactivity during the stagnation period (A) from May 17th – May 22nd in 2016 and the transport period (B) from 28 May – 1 June 2016. The measured and calculated OH reactivity are on the left. The blue shading represents an uncertainty of 16.7% at 1σ. The black bars represent the propagated uncertainty of 20.1% at 1σ from calculated missing OH reactivity. The percent missing OH reactivity is on the right axis (the figure caption in the revised manuscript)

---

## Author Response (AR2)

This is a review of the REVISED MANUSCRIPT performed by Reviewer #3.
All line numbers given below are for the revised version of the manuscript.

The authors have provided additional information that answers most of the comments from the 3 reviewers. However, there are still a few points that need to be addressed before publication in ACP:

We appreciate Reviewer 3's insightful and constructive comments, which significantly improve the manuscript. Please find our response as shown below.

1/ The authors used a catalytic converter to generate zero air from ambient air. The term "VOC-free air" is more appropriate since ambient NOx are not removed. This zero air is used to perform the "C2" measurement. NOx will be present in the CRM reactor during both the "C3" measurement, when ambient air is sampled, but also during the "C2" measurement. Since OH can react with NO and NO2 during both "C3" and "C2", the CRM should be blind to the OH reactivity generated by NOx. The reported measurements of OH reactivity may therefore be biased low. Interestingly, the authors indicated L185-186 that during the SOAS campaign measurements from the UCI CRM instrument were 16% lower on average than measurements from a LIF system. Could the authors comment on this?

We have discussed the possibility and accept the limitation of this study.
Now it reads:

In conclusion, it is possible that our reported OH reactivity may systematically underestimate ambient total OH reactivity as much as ambient OH reactivity coming from $NO_2$. (Line 206 - 208, In the track change version)

2/ The authors provided more information about operating conditions for their CRM instrument and indicated that the pyrrole-to-OH ratio was kept constant at a value of 3. This ratio can depend on the amount of OH that is produced from the photolysis of ambient water-vapor in the reactor due to the leakage of 185-nm photons. As a consequence the ratio usually changes with ambient humidity. This ratio was found to vary significantly for other CRM instruments when operated continuously during field campaigns. How did the authors manage to keep the pyrrole-to-OH constant? Was the geometry of the CRM reactor optimized to avoid the photolysis of ambient water-vapor?

We've realized that even with As we are adding 150 cc per minute of humidified N2 (likely 100 % RH) to the reactor with flow rate of 240 cc per minute, therefore, the variation of ambient humidity change would be dampened quite a bit from the mixing effect. We have included the discussion and now it reads:

Even in the field environment with various relative humidity, we have not observed noticeable changes in this ratio as we flow bulk humidified nitrogen (150 sccm) to the reactor with the total flow of 240 cc, which result in dampening the temporal ambient relative humidity variations. (Line 210 - 213, in the track change version)

Minor comments:

L192-193: "An extensive intercomparison study was conducted by Fuchs et al. (2017) with various OH reactivity measurement techniques that highlighted potential analytical artifacts in the CRM technique. These artifacts have all been examined and preventive measures have been implemented in the UCI CIMS-CRM system deployed at TRF." - I would add some caveat here since these artefacts have not been fully investigated for this instrument. While some testing has been performed to check whether ambient NO and O3 could lead to measurement artifacts, the authors acknowledged in their responses to the first review that additional tests are needed to well characterize these artefacts.

We have clarified the limitation in the revised manuscript. Now, it reads:

Again, as the CRM method is relatively new technique, one should keep in mind that the future studies may find potential artifacts that we do not report in this study. (Line 223 - 225, in the track change version)

L200-201: "Our approach to this type of interference has been to determine the maximum NO level, noticeably interfering with the calibration regression line shown in Sanchez et al. (2018). Laboratory tests indicate that the statistical agreement started to veer off when the NO level is 5 ppb in 1 $\sigma$ of the linear regression" – These tests are of interest for the CRM community and the reviewer recommends to show them in the supplementary material. This will also provide additional confidence in the dataset.

We have added the quantitative information. Now it reads.

Laboratory tests indicate that the statistical agreement started to veer off when the NO level is 5 ppb in 1  of the linear regression as the slope for the calibration curve has changed from 0.238 to 0.246. (Line 201 - 203, in the track change version)

L214-216: "In the 2015 field campaigns conducted in Seoul South Korea (Kim et al., 2016), we conducted a standard addition experiment for the propene standard for additional ~ 30 s-1 in two different ozone environment 65 ppb and 123 ppb. The outcome illustrates an agreement between two additions within the analytical uncertainty." – While a standard addition test could highlight an artefact impacting the linear response of CRM to OH reactivity, it cannot rule out an artefact leading to a positive or negative offset that would only depends on O3. For the later, a standard addition of 30s-1 of OH reactivity would always lead to the right change in the measured total OH reactivity. However, the total OH reactivity with (ambient reactivity + standard addition reactivity) or without (ambient reactivity) standard addition would be biased low by the same amount. This should be acknowledge in the manuscript.

We have clarified the limitation in the revised manuscript. Now, it reads:

The outcome illustrates an agreement between two additions within the analytical uncertainty although a systematic laboratory study will warrant an accurate uncertainty assessment from ozone. (Line 222 - 224, in the track change version)

L206-211: "Therefore, we performed multi-point calibrations with a propene mixture using a NIST traceable gas standard (AirLiquide LLC, 0.847 ppm) during the field campaign to avoid any circumstances where the pseudo first-order reaction regime is not established." – Please indicate the range of OH reactivity generated during the multipoint calibration.

We have included the information. Now, it reads:

we performed multi-point calibrations (5 $s^{-1}$ to 30 $s^{-1}$) with a propene mixture using a NIST traceable gas standard (AirLiquide LLC, 0.847 ppm) during the field campaign to avoid any circumstances where the pseudo first-order reaction regime is not established. (Line 214 - 217, in the track change version)

L244-245: The writing of Eq. 1 is still confusing (mixing of parameters and units). Please only use parameters that are defined in the main text. For instance:
MRvoc=Svoc*kbenzene/kvoc*1/Rbenzene.
MR:Mixing Ratio, Svoc: normalized voc signal, kbenzene and kvoc: proton transfer rate constants, Rbenzene: benzene sensitivity

We have revised the equation as suggested. Now it reads.

[revised manuscript text omitted]

---

## Author Response (AR3)

We appreciate constructive comments from Dr. Hofzumahaus. We have revised and submitted the manuscript. The specifics of this revision are described as below

The sentence (lines 206-208) added in response to the referee's comment 1 is misleading in connection with the preceding sentence. In the paper by Fuchs et al. (2017), different OH reactivity measurement instruments were compared and possible interferences due to OH recyling in the instruments in the presence of NOx were discussed. However, Referee #3 has addressed a completely different problem which is caused by the use of a catalytic converter to generate zero air from ambient air. If the catalytic converter does not remove NO2, then the signal difference C3-C2 of the CRM instrument is not sensitive to ambient NO2 and the measured ambient OH reactivity will be underestimated by the amount of NO2 reactivity. This explanation needs to be added for clarification. The resulting bias should also be discussed in Section 3, when the measured OH reactivity is compared to the reactivity calculated from the measured compounds in Table 1.

We agree that the previous revision can be mis-leading. We have revised manuscript to reflect this comment. Now, the manuscript reads:

Line 207 – 209 . In conclusion, one should note that our reported OH reactivity could potentially underestimate actual ambient OH reactivity as much as the contributions from those from ambient $NO_2$.

Line 351 – 353, . Again, the potential underestimation in ambient OH reactivity as much as the contributions from ambient $NO_2$, presented in the method section, should be well noted.

The last point by Referee #3 adressing Eq. 1 is not satisfactorily corrected. The revised equation is still mixing symbols and units. The physical quantities (mixing ratio, count rate, detection sensitivity) should be represented by symbols defined in the text.

We have revised the equation as the current form still may cause confusion. Now it read in the trackchange version line 257 -268

[revised manuscript text omitted]